# Genetic loci and metabolic states associated with murine epigenetic aging

**Khyobeni Mozhui[1,2]\*, Ake T Lu[3], Caesar Z Li[3], Amin Haghani[4], Jose Vladimir Sandoval-Sierra[1], Yibo Wu[5,6], Robert W Williams[2], Steve Horvath[3,4]**

[1]Department of Preventive Medicine, University of Tennessee Health Science Center, College of Medicine, Memphis, United States; [2]Department of Genetics, Genomics and Informatics, University of Tennessee Health Science Center, College of Medicine, Memphis, United States; [3]Department of Human Genetics, David Geffen School of Medicine, University of California Los Angeles, Los Angeles, United States; [4]Department of Biostatistics, Fielding School of Public Health, University of California Los Angeles, Los Angeles, United States; [5]YCI Laboratory for Next-Generation Proteomics, RIKEN Center for Integrative Medical Sciences, Yokohama, Japan; [6]University of Geneva, Geneva, Switzerland

**\*For correspondence:**
kmozhui@uthsc.edu

**Abstract** Changes in DNA methylation (DNAm) are linked to aging. Here, we profile highly conserved CpGs in 339 predominantly female mice belonging to the BXD family for which we have deep longevity and genomic data. We use a 'pan-mammalian' microarray that provides a common platform for assaying the methylome across mammalian clades. We computed epigenetic clocks and tested associations with DNAm entropy, diet, weight, metabolic traits, and genetic variation. We describe the multifactorial variance of methylation at these CpGs and show that high-fat diet augments the age-related changes. Entropy increases with age. The progression to disorder, particularly at CpGs that gain methylation over time, was predictive of genotype-dependent life expectancy. The longer-lived BXD strains had comparatively lower entropy at a given age. We identified two genetic loci that modulate epigenetic age acceleration (EAA): one on chromosome (Chr) 11 that encompasses the *Erbb2/Her2* oncogenic region, and the other on Chr19 that contains a cytochrome P450 cluster. Both loci harbor genes associated with EAA in humans, including *STXBP4*, *NKX2*-3, and *CUTC*. Transcriptome and proteome analyses revealed correlations with oxidation-reduction, metabolic, and immune response pathways. Our results highlight concordant loci for EAA in humans and mice, and demonstrate a tight coupling between the metabolic state and epigenetic aging.

## Editor's evaluation

This article used three newly generated epigenetic predictors to test how they differ between genetically diverse mice from the BXD family (by looking at metabolic traits and lifespan). The authors subsequently identified several quantitative trait loci for the different predictors, using linkage analysis, and performed transcriptome and proteome analyses of liver and adipose tissue. The described results provide some important new insights on the underlying biology of epigenetic mouse aging and may be used to inform future studies in other model organisms and humans focused on studying the relationship between epigenetic aging and metabolism.

## Introduction

Epigenetic clocks are widely used molecular biomarkers of aging (*Horvath and Raj, 2018*). These DNA methylation (DNAm) age predictors are based on the methylation levels of select CpGs that are

distributed across the genome. Each CpG that is used in a clock model is assigned a specific weight, typically derived from supervised training algorithms (*Bell et al., 2019*; *Thompson et al., 2018*; *Porter et al., 2021*), and collectively, the methylation status across this ensemble of 'clock CpGs' is used to estimate the epigenetic age (DNAmAge). This estimate tracks closely, but not perfectly, with an individual's chronological age. How much the DNAmAge deviates from the known chronological age can be a measure of the rate of biological aging. Denoted as 'epigenetic age acceleration' (EAA), a more accelerated clock (positive EAA) suggests an older biological age, and a decelerated clock (negative EAA) suggests a younger biological age. The extent of age acceleration has been associated with variation in health, fitness, exposure to stressors, body mass index (BMI), and even life expectancy (*Zannas et al., 2015*; *Marioni et al., 2015*; *Dugué et al., 2018*; *Lu et al., 2019*; *Ryan et al., 2020*).

DNAm clocks were initially reported for humans (*Hannum et al., 2013*; *Horvath, 2013*; *Bocklandt et al., 2011*). Since then, many different models of human DNAmAge have been developed, and this rapid expansion was made possible by reliable DNAm microarrays that provide a fixed CpG content – starting with the Illumina Infinium 27K to the current 850K EPIC array (*Horvath, 2013*; *Zhang et al., 2019*; *Liu et al., 2020*; *Lee et al., 2020*). These clock variants differ in the subset of CpGs that go into the age estimation model. Some clock models are specific to cells or tissues, others are multitissue. Some clocks perform better at predicting chronological age, others better capture biological aging and predict health and life expectancy (*Lu et al., 2019*; *Horvath et al., 2018*; *Levine et al., 2018*; *Shireby, 2020*). The performance of these clocks depends heavily on the training models, and the size and tissue types of the training set (*Zhang et al., 2019*).

The DNAm age biomarker has also been extended to model organisms, and this has opened up the possibility of directly testing the effects of different interventions such as caloric restriction, rapamycin, and genetic manipulation (*Thompson et al., 2018*; *Petkovich, 2017*; *Stubbs et al., 2017*; *Wang, 2017*; *Levine et al., 2020*; *Han, 2018*). However, one point to note is that model organisms have not benefited from a microarray platform comparable to that of the human methylation Infinium arrays. Most rodent studies have used enrichment-based DNAm sequencing, and this limits the transferability and reproducibility of clocks between datasets since the same CpGs are not always covered (*Wang, 2017*). Moreover, these studies are usually performed in a single inbred strain (for mouse, the canonical C57BL/6), or at most, a few genetic backgrounds, and this makes it impossible to carry out genetic mapping studies that can complement the human genome-wide association studies (GWAS) of epigenetic aging (*Gibson, 2019*; *Lin, 2021*; *Lu et al., 2018*; *McCartney et al., 2021*; *Kuo et al., 2021*).

A new microarray was recently developed to profile CpGs that have high conservation in mammals. This pan-mammalian DNAm array (HorvathMammalMethylChip40) surveys over 37K CpGs and provides a unifying platform to study epigenetic aging in mammals (*Arneson et al., 2022*). This array has been used to build multitissue universal clocks and lifespan predictors that are applicable to a variety of mammalian species (*Lu et al., 2021*; *Li et al., 2021*). Here, we use this array to examine the dynamism and variability of the conserved CpGs in a genetically diverse cohort of mice belonging to the BXD family (*Williams et al., 2022*; *Roy et al., 2021*).

The BXDs are one of the preeminent murine genetic reference panels used as the experimental paradigm of precision medicine (*Ashbrook et al., 2021*). They are a large family of recombinant inbred (RI) strains made by crossing the C57BL/6J (B6) and DBA/2J (D2) parental strains. The family has been expanded to 150 fully sequenced progeny strains (*Ashbrook et al., 2021*; *Peirce et al., 2004*). The individual members of the BXD family (e.g., BXD1, BXD27, BXD102) represent a replicable isogenic cohort. The family segregates for a high level of genetic variation, and likewise, family members have high variation in their metabolic profiles, responses to diet, aging rates, and life expectancies (*Williams et al., 2022*; *Roy et al., 2021*; *Ashbrook et al., 2021*; *de Haan and Williams, 2005*; *Hsu et al., 2003*; *Lang et al., 2010*). The availability of deep genome sequence data, and unrivaled multi-omic and phenomic data make the BXDs a powerful tool with which to evaluate the causal linkage between genome, epigenome, and aging rates.

In our previous work, we used an enrichment-based sequencing to assay the methylome in a modest number of BXD mice and reported rapid age-dependent methylation changes in mice on high-fat diet (HFD) and mice with higher body weight (*Sandoval-Sierra et al., 2020*). In this study, we start by testing the performance of new pan-tissue and liver-specific epigenetic mouse clocks and evaluate how these relate to metabolic states, genotype-dependent life expectancy, and methylome entropy.

We also apply a multifactor analysis of site-specific CpG methylation to understand the interrelations among four key variables – chronological age, diet, weight, and lifespan – and the liver methylome. We perform quantitative trait locus (QTL) mapping, along with multi-omic gene expression analyses, and identify upstream gene loci that modulate the DNAm clocks.

Our results are consistent with a faster clock for cases on HFD and with higher body weight. This may be partly because exposure to HFD augmented the age-dependent gains in methylation at specific CpGs. We also observed that BXD genotypes with longer life expectancies tend to have lower methylation at CpGs that undergo age-dependent methylation gains, and the entropy computed from this set of CpGs has a significant inverse correlation with strain lifespan. QTL mapping uncovered loci on chromosomes (Chrs) 11 and 19 that are associated with EAA. A strong candidate gene in the chromosome (Chr) 11 interval (referred to as Eaa11) is *Stxbp4*, a gene that has been consistently linked to EAA by human GWAS (*Gibson, 2019*; *Lu et al., 2018*; *McCartney et al., 2021*). The Chr19 QTL (Eaa19) also harbors strong contenders including *Cyp26a1*, *Myof*, *Cutc*, and *Nkx2–3*, and the conserved genes in humans have been associated with longevity and EAA (*McCartney et al., 2021*; *Benton et al., 2017*; *Yashin et al., 2018*). We performed gene expression analyses using transcriptomic and proteomic data to clarify the molecular pathways linked to epigenetic aging, and this highlighted metabolic networks, and also apolipoproteins (including APOE) as strong expression correlates.

## Results

### Description of samples

Liver DNAm data was from 321 females and 18 males belonging to 45 members of the BXD family, including both parental strains and F1 hybrids. Age ranged from 5.6 to 33.4 months. Mice were all weaned onto a normal chow (control diet [CD]), and a balanced subset of cases were then randomly assigned to HFD (see *Roy et al., 2021* for details). Tissues were collected at approximately 6 months' intervals (see *Williams et al., 2022*). Individual-level data are given in *Supplementary file 1*.

### DNAm clocks, entropy, and chronological age prediction

We built three different clocks, and each was developed as a pair depending on whether the training set used all tissues (pan-tissue) or a specific tissue (in this case, liver). These are (1) a general DNAm clock (referred to simply as DNAmAge): clock trained without pre-selecting for any specific CpGs; (2) developmental clock (dev.DNAmAge): built from CpGs that change during development (defined as the period from prenatal to 1.6 months); and (3) interventional clock (int.DNAmAge): built from CpGs that change in response to aging-related interventions (caloric restriction and growth hormone receptor knockout). The clocks we report here were trained in a larger mouse dataset that excluded the BXDs and are therefore unbiased to the characteristics of the BXD family (*Lu et al., 2021*; *Li et al., 2021*; *Haghani et al., 2022*). The specific clock CpGs and coefficients for DNAmAge computation are given in *Supplementary file 2*. All clocks performed well in age estimation and had an average r of 0.89 with chronological age. However, the interventional clocks had higher deviation from

**Table 1.** Chronological age prediction and correlation with methylome-wide entropy.

| Clock type | DNAmAge name | Tissue | r with age (n = 339)* | Age prediction median error | r with entropy (n = 339)*, † |
|---|---|---|---|---|---|
| | | Pan | 0.89 | 0.12 | 0.43 |
| Standard clocks | DNAmAge | Liver | 0.92 | 0.10 | 0.40 |
| | | Pan | 0.87 | 0.14 | 0.39 |
| Developmental clocks | dev.DNAmAge | Liver | 0.91 | 0.12 | 0.37 |
| | | Pan | 0.85 | 0.17 | 0.29 |
| Interventional clocks | int.DNAmAge | Liver | 0.86 | 0.15 | 0.47 |

*p<0.0001.
†Methylome-wide entropy calculated from ~28K CpGs.

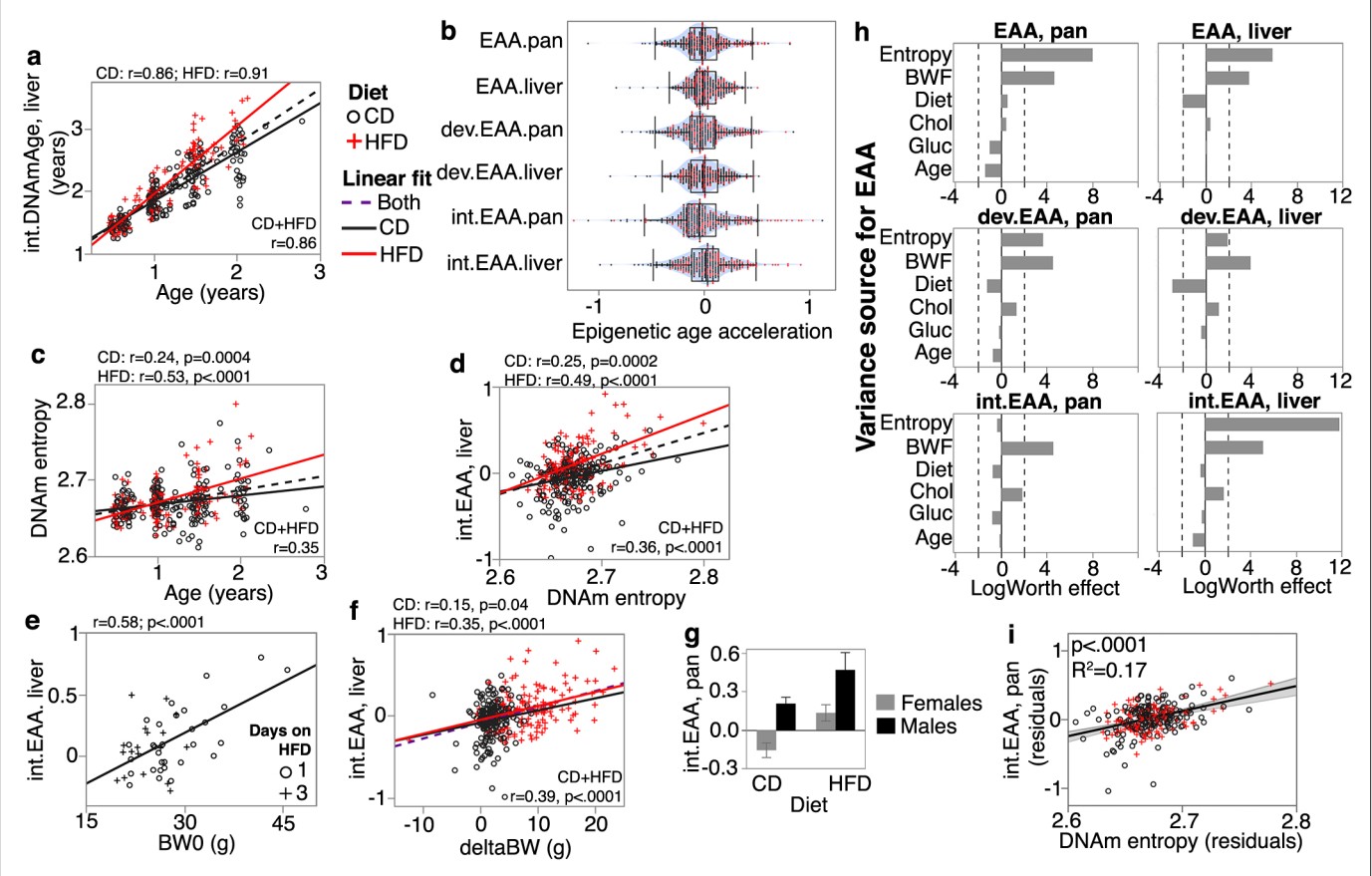

**Figure 1.** Correlates and modifiers of epigenetic clocks and methylome-wide entropy. (**a**) Correlation between chronological age and predicted age (shown for the liver intervention clock or int.DNAmAge). Black circles are control diet (CD, n = 210); red crosses are high-fat diet mice (HFD, n = 128). (**b**) Violin plots of age-adjusted epigenetic age acceleration (EAA) ('int,' interventional; 'dev,' developmental). (**c**) Shannon entropy, calculated from the full set of high-quality CpGs, increases with age. (**d**) Methylome entropy has a direct correlation with EAA (shown for the liver int.EAA). (**e**) For 48 mice, initial body weight (BW0) was measured 1 or 3 days after introduction to HFD, and these showed significant correlation with EAA. (**f**) Weight was first measured at mean age of 4.5 ± 2.7 months (BW0), and then at 6.3 ± 2.8 months (BW1). Weight gains during this interval (deltaBW = BW1 – BW0) are a direct correlate of EAA. (**g**) For BXD genotypes with males and females samples, males have higher age acceleration. Bars represent mean ± standard error; 40 females (26 CD, 14 HFD) and 18 males (10 CD, 8 HFD). (**h**) Relative effects of different predictor variables on EAA shown as logworth scores (-log$_{10}$p). The dashed lines correspond to p=0.01. Positive logworth values indicate positive regression estimates (for diet, positive means higher in HFD compared to CD). BWF, final weight; Chol, serum total cholesterol; Gluc, fasted glucose levels. (**i**) The residual plot displays association between methylome-wide entropy and the pan-tissue int.EAA after adjustment for diet, age, weight, glucose, cholesterol, and batch.

The online version of this article includes the following figure supplement(s) for figure 1:

**Figure supplement 1.** Relative effects of different predictor variables on epigenetic age acceleration (EAA).

**Figure supplement 2.** BXD strains with shorter life expectancy have slightly more accelerated clocks.

chronological age and higher median predictive error (*Table 1*, *Figure 1a*). The age-adjusted EAA derived from these clocks showed wide individual variation (*Figure 1b*).

We next estimated the methylome-wide entropy as a measure of randomness and information loss. This was computed from 27,966 probes that provide high-quality data and have been validated to perform well in mice (*Arneson et al., 2022*). Consistent with previous reports (*Hannum et al., 2013*; *Sziráki et al., 2018*), this property increased with chronological age, and age accounted for about 6% (in CD) to 28% (in HFD) of the variance in entropy (*Figure 1c*). As direct correlates of chronological age, all the DNAmAge estimates also had significant positive correlations with entropy (*Table 1*). We hypothesized that higher entropy levels will be associated with higher EAA, and based on this bivariate comparison, most of the EAA showed a significant positive correlation with entropy (*Figure 1d*; *Supplementary file 3*).

**Table 2.** Association with diet and weight, and heritability of the epigenetic readouts.

| Type | EAA | Diet | Mean (SD) | Diet (p) | r BW0* | p BW0 | r BWF* | p BWF | h² | Strain r† |
|---|---|---|---|---|---|---|---|---|---|---|
| | EAA, pan | CD | –0.05 ± 0.21 | | 0.19 | 0.006 | 0.29 | <0.0001 | 0.49 | |
| | | HFD | 0.07 ± 0.21 | <0.0001 | 0.21 | 0.01 | 0.42 | <0.0001 | 0.50 | 0.54 |
| | EAA, liver | CD | 0 ± 0.17 | | 0.09 | ns | 0.20 | 0.003 | 0.40 | |
| | | HFD | 0.03 ± 0.14 | ns | 0.22 | 0.01 | 0.49 | <0.0001 | 0.52 | 0.73 |
| | dev.EAA, pan | CD | –0.04 ± 0.23 | | 0.09 | ns | 0.22 | 0.001 | 0.53 | |
| | | HFD | 0.03 ± 0.22 | 0.004 | 0.27 | 0.002 | 0.45 | <0.0001 | 0.61 | 0.76 |
| | dev.EAA, liver | CD | 0 ± 0.2 | | 0.19 | 0.002 | 0.29 | <0.0001 | 0.46 | |
| | | HFD | 0 ± 0.16 | ns | 0.29 | 0.0007 | 0.47 | <0.0001 | 0.60 | 0.78 |
| | int.EAA, pan | CD | –0.05 ± 0.25 | | 0.03 | ns | 0.21 | 0.002 | 0.27 | |
| | | HFD | 0.06 ± 0.33 | 0.0003 | 0.22 | 0.01 | 0.46 | <0.0001 | 0.45 | 0.66 |
| Mouse clocks | int.EAA, liver | CD | –0.04 ± 0.22 | | 0.05 | ns | 0.18 | 0.01 | 0.59 | |
| | | HFD | 0.11 ± 0.25 | <0.0001 | 0.27 | 0.002 | 0.58 | <0.0001 | 0.54 | 0.80 |
| | | CD | 2.67 ± 0.02 | | 0.09 | ns | 0.05 | ns | 0.31 | |
| Entropy | - | HFD | 2.67 ± 0.02 | ns | 0.15 | 0.09 | 0.15 | 0.09 | 0.32 | 0.24 (ns) |

CD, control diet; HFD, high-fat diet; EAA, epigenetic age acceleration; int, interventional; dev, developmental.
*BW0 is body weight at about 4.5 months of age (n = 339; 210 CD and 129 HFD); BWF is final weight at tissue collection (1 HFD case missing data; n = 338; 210 CD and 128 HFD).
†Pearson correlation between strain means for n = 29 BXD genotypes kept on CD and HFD.

## How the epigenetic readouts relate to diet, sex, and metabolic traits
### Diet
HFD was associated with higher EAA for four of the clocks (*Table 2*). For instance, the liver-specific interventional clock diverged between the diets (*Figure 1a*), and CD mice had an average of –0.04 years of age deceleration, and HFD mice had an average of +0.11 years of age acceleration (*Table 2*). The two clocks that were not affected by diet were the liver general and developmental clocks. Methylome-wide entropy was not different between the diets.

### Body weight
Body weight was first measured when mice were at an average age of 4.5 ± 2.7 months. We refer to this initial weight as baseline body weight (BW0). For mice on HFD, this was usually before introduction to the diet, except for 48 cases that were first weighed 1 or 3 days after HFD (*Supplementary file 1*). In the CD group, only the EAA from the pan-tissue standard and liver developmental clocks showed modest but significant positive correlations with BW0 (*Table 2*). In the HFD group, the positive correlation with BW0 was more robust and consistent across all the clocks, and this may have been due to the inclusion of the 48 cases that had been on HFD for 1 or 3 days. Taking only these 48 cases, we found that higher weight even after 1 day of HFD had an age-accelerating effect on all the clocks, and this was particularly strong for the interventional clocks ($r = 0.45$, p=0.001 for the pan-tissue int.EAA; $r = 0.58$, p<0.0001 for the liver int.EAA) (*Figure 1e*). Second weight was measured 7.4 ± 5.2 weeks after BW0 (mean age 6.3 ± 2.8 months). We refer to this as BW1 and estimated the weight change as deltaBW = BW1 – BW0. DeltaBW was a positive correlate of EAA in both diets, albeit more pronounced in the HFD group (*Figure 1f*). The final body weight (BWF) was measured at the time of tissue harvest, and EAA from all the mouse clocks were significant correlates of BWF in both diets (*Table 2*). In contrast, entropy did not show an association with either BW0 or BWF. We do note that when stratified by diet the entropy level had a slight positive correlation with BW1 in the HFD group ($r = 0.23$, p=0.008), but not in the CD group (*Supplementary file 3*).

## Sex

Four BXD genotypes (B6D2F1, D2B6F1, BXD102, B6) had cases from both males and females. We used these to test for sex effect. All the clocks showed significant age acceleration in male mice, and this was particularly strong for both dev.EAA and pan-tissue int.EAA (*Figure 1g*, *Supplementary file 3*). The effect of sex was independent of the higher BWF of males, and the higher age acceleration in males was detected after adjusting for BWF (*Supplementary file 4a*). There was no significant difference in entropy between the sexes.

## Metabolic measures

278 cases with DNAm data also had fasted serum glucose and total cholesterol (*Williams et al., 2022*; *Roy et al., 2021*), and we examined whether these metabolic traits were associated with either the EAA measures or methylome entropy. Since these are highly dependent on diet, weight, and age, we applied a multivariable model to jointly examine how the different metabolic variables (cholesterol and glucose, as well as diet and weight) and entropy relate to EAA after adjusting for age. To test the robustness of associations, we also include the methylation array batch as another covariate (*Supplementary file 1* has batch information; *Supplementary file 5* has the full statistics). *Figure 1h* shows the relative strengths and directions of associations between these variables and the EAA traits. Except for the pan-tissue interventional clock, entropy had a strong positive correlation with EAA. For example, a plot of residuals between entropy and the liver int.EAA indicates that after adjusting for all the other covariates, the methylome-wide entropy explains 17% of the variance in int.EAA (*Figure 1i*). Since diet strongly influences BWF, the inclusion of BWF in the regression diminished the effect of diet. For the two clocks that were not influenced by diet (the liver EAA and liver dev.EAA), adjusting for the effect of BWF resulted in an inverse association with diet (i.e., the residual EAA values after accounting for BWF were slightly lower in the HFD group). Fasted glucose did not have a significant effect on EAA. Cholesterol had a positive association with the interventional clocks but the effects were modest (residual $R^2 = 0.02$ and p=0.02 for cholesterol and the pan-tissue int.EAA).

We also performed a similar analysis with BW0 instead of BWF (*Figure 1—figure supplement 1*), and here, HFD remained as an accelerator of the pan-tissue EAA and liver int.EAA. Cholesterol also became a significant positive correlate of EAA for the interventional clocks. This would suggest that the effect of diet on EAA is mostly mediated by its impact on physiological and metabolic traits, and BWF becomes a prominent predictor of EAA.

To summarize, our results indicate that the degree of disorder in the methylome increases with age and may partly contribute to the epigenetic clocks as higher entropy is associated with higher EAA. The EAA traits were also associated with biological variables (i.e., body weight, diet, and sex). Of these, BWF was the strongest modulator of EAA.

## How the epigenetic readouts relate to strain longevity

We next obtained longevity data from a parallel cohort of female BXD mice that were allowed to age on CD or HFD (*Roy et al., 2021*). Since the strain lifespan was determined from female BXDs, we restricted this to only the female cases. For strains with natural death data from n ≥ 5, we computed the minimum (minLS), 25th quartile (25Q-LS), mean, median lifespan, 75th quartile (75Q-LS), and maximum lifespan (maxLS) (*Supplementary file 1*). Specifically, we postulated an accelerated clock for strains with shorter lifespan. Overall, the EAA measures showed the expected inverse correlation trend with the lifespan statistics (*Supplementary file 4b*). However, these correlations were weak. The correlations were significant only for the pan-tissue general clock (*Figure 1—figure supplement 2a*) and the liver intervention clock, with explained variance in lifespan of only ~3% (*Figure 1—figure supplement 2b and c*). When separated by diet, these correlations became weaker, indicating that while we see the expected inverse relationship, the EAA is only weakly predictive of strain longevity. Entropy estimated from the full set of CpGs was unrelated to strain longevity.

## Multifactor variance of the conserved CpGs

Both entropy and clock readouts capture the overall variation across multiple CpGs, and to gain insights into the underlying variance patterns, we performed a multivariable epigenome-wide association study (EWAS). For this, we applied a site-by-site regression on the 27,966 validated CpGs

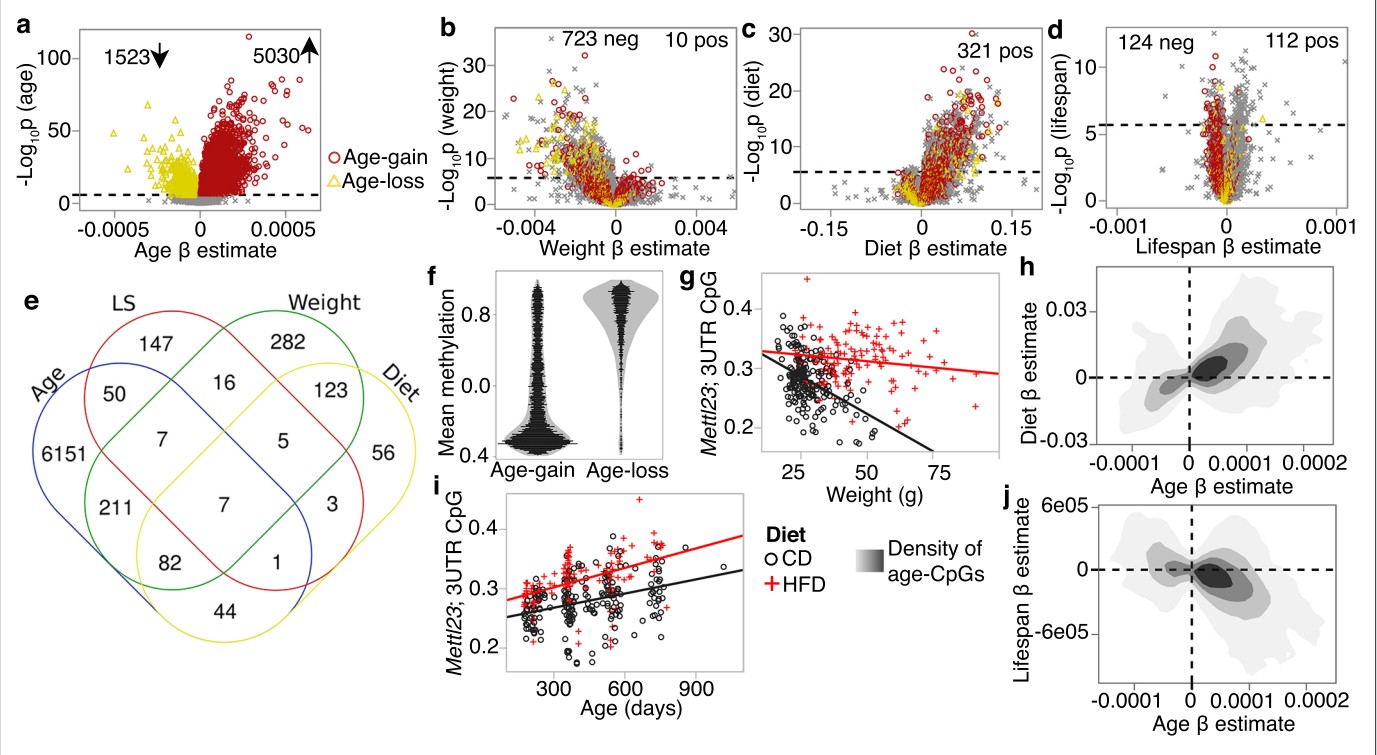

**Figure 2.** Multivariable analysis of site-specific methylation. (**a**) Volcano plot comparing regression estimates (change in methylation beta-value per day of age) versus the statistical significance for age effect. Dashed line denotes the Bonferroni p=0.05 for ~28K tests. Similar volcano plots for predictor variables: (**b**) final body weight (regression estimates are change per gram of weight), (**c**) diet (change in high-fat diet [HFD] compared to control diet [CD]), and (**d**) strain median lifespan (per day increase in median longevity). CpGs that were significantly associated with age are denoted by colored markers (red circles: age-gain; yellow triangles: age-loss). (**e**) Overlap among the lists of differentially methylated CpGs. (**f**) Each dot represents the mean methylation beta-values for the 5030 age-gain and 1523 age-loss CpGs. (**g**) Correlation between body weight and methylation beta-values for the CpG (cg10587537) located in the 3'UTR of *Mettl23*. Mice on HFD have higher methylation than mice on CD, but the inverse correlation with weight is consistent for both groups (r = –0.45, p<0.0001 for CD; r = –0.15, p=0.08 for HFD). (**h**) Contour density plot for the 6553 CpGs that are significantly associated with age (age-DMCs). This relates the pattern of change with age (x-axis) with change on HFD (y-axis). CpGs that gain methylation with age are also increased in methylation by HFD. (**i**) Correlation between age and methylation at the *Mettl23* 3'UTR CpG (r = 0.35 for CD; r = 0.46 for HFD). (**j**) For the 6553 age-DMCs, the contour density plot relates the pattern of change with age (x-axis) vs. change with median longevity (y-axis). CpGs that gain methylation with age have lower methylation with higher lifespan.

The online version of this article includes the following figure supplement(s) for figure 2:

**Figure supplement 1.** Genomic and chromatin states of differentially methylated CpGs.

**Figure supplement 2.** Array quality check.

(***Arneson et al., 2022***) and tested for association with age, BWF, diet, and genotype-dependent strain median lifespan (full set of probes, annotations, and EWAS results in ***Supplementary file 6***).

Age was clearly the most influential variable, and this is apparent from the volcano plots (***Figure 2a–d***). We used a cutoff of Bonferroni p≤0.05 to define differentially methylated CpGs (DMCs), and 6553 CpGs were associated with age (referred to as age-DMCs), 733 with weight (weight-DMCs), 321 with diet (diet-DMCs), and 236 with genotype-dependent lifespan (LS-DMCs). We note extensive overlap among the lists of DMCs that shows that variation at these CpGs are multifactorial in nature (***Figure 1e***). Majority of the age-DMCs (77%) gained methylation (or age-gain), and consistent with previous observations, age-gain CpGs tended to be in regions with low methylation, whereas age-DMCs that declined in methylation (age-loss) were in regions with high methylation (***Figure 2f***; ***Sandoval-Sierra et al., 2020***; ***Sziráki et al., 2018***; ***Slieker et al., 2016***). By overlaying the volcano plots with the age-gain and age-loss information, we see distinct patterns in how these age-DMCs vary with weight (***Figure 2b***), diet (***Figure 2c***), and genotype lifespan (***Figure 2d***). While the majority of CpGs, including several age-loss CpGs, had negative regression estimates for weight (i.e., decrease in DNAm with unit increase in weight), HFD was associated with higher methylation levels (positive

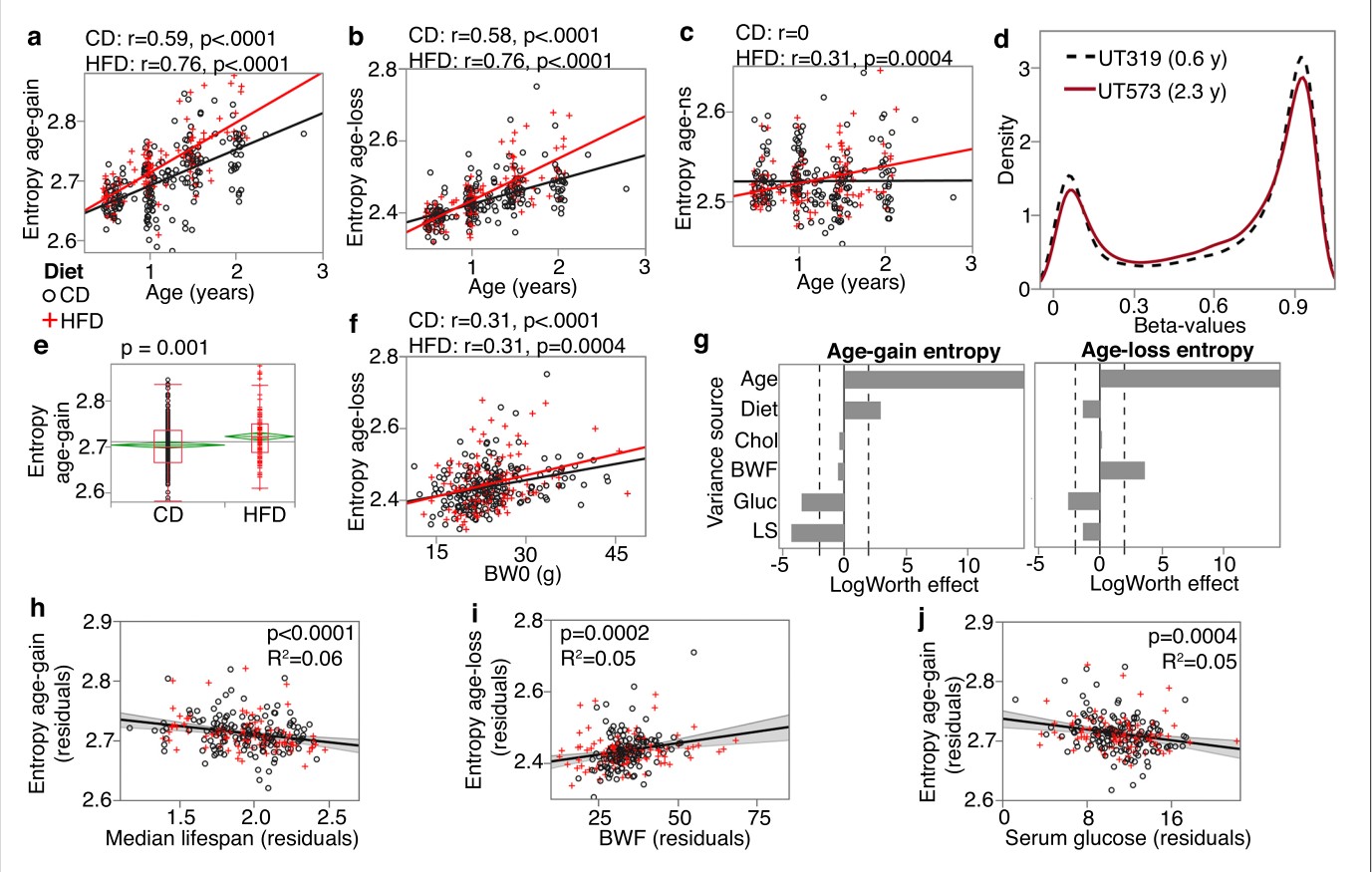

**Figure 3.** Entropy at age-associated CpGs. Entropy values were calculated for the 5020 age-gain and 1523 age-loss CpGs separately. For both control diet (CD) and high-fat diet (HFD), there is a significant increase in entropy with age at the (**a**) age-gain and (**b**) age-loss CpGs. (**c**) The HFD mice also showed a slight increase in entropy at CpGs that were not strongly associated with age (age-ns). (**d**) The methylome-wide distribution of beta-values in a young adult mouse (0.6 years old; black dashed line) and an older mouse (2.3 years old; red line); both CD mice. The young mouse has higher peaks at the hypomethylated (closer to 0.1) and hypermethylated (around 0.9) beta-values compared to the older mouse. (**e**) The HFD group has higher entropy at the age-gain CpGs compared to the CD group. (**f**) Entropy at age-loss CpGs is higher with higher baseline weight (BW0). (**g**) Relative effects of predictor variables on entropy shown as logworth scores (-log₁₀p). The dashed lines correspond to p=0.01. Positive values indicate positive regression estimates (for diet, positive value means higher in HFD). BWF, final weight; Chol, serum total cholesterol; Gluc, fasted glucose levels; LS, strain median lifespan. (**h**) The residual plot (adjusted for age, diet, BWF, glucose, cholesterol, and batch) shows the inverse association between entropy at age-gain sites and lifespan. Similar residual plots show the association between (**i**) BWF and age-loss entropy, and (**j**) between fasted serum glucose and age-gain entropy.

regression estimates) including at several age-DMCs (**Figure 2c**). This pattern of inverse correlation with weight but heightened methylation due to HFD is illustrated by the CpG in the 3'UTR of *Mettl23* (cg10587537) (**Figure 2g**). Taking the 6553 age-DMCs, a comparison of the regression estimates for age (i.e., the change in methylation per day of aging) versus diet (difference in HFD relative to CD) shows that the age-gains were augmented in methylation by HFD (**Figure 2h**), and again, this is illustrated by the CpG in the *Mettl23* 3'UTR (**Figure 2i**). For the LS-DMCs, sites that had negative regression estimates for lifespan (i.e., lower DNAm per day increase in strain median longevity) had higher proportion of age-gain CpGs (**Figure 2d**). A comparison between the regression estimates for age versus the regression estimates for lifespan shows that CpGs that gain methylation with age tended to have lower methylation in strains with longer lifespan (**Figure 2j**).

As in *Sziráki et al., 2018*, we divided the CpGs by age effect: age-gain, age-loss, and those that do not change strongly with age (age-ns; i.e., the remaining 21,413 CpGs that were not classified as age-DMCs). For these conserved CpGs, both sets of age-DMCs had significant increases in entropy with age regardless of diet (**Figure 3a and b**), and even the age-ns showed a modest entropy gain with age in the HFD group (**Figure 3c**). The reason for this increase in disorder becomes evident when we compare the density plots using the full set of CpGs for one of the younger mice (UT319;

0.56 years old) and one of the older mice (UT573; 2.3 years) (*Figure 3d*). Concordant with previous reports (*Sziráki et al., 2018*; *Kerepesi et al., 2022*), the older sample showed a subtle flattening of the bimodal peaks towards a slightly more hemi-methylated state. The entropy of the age-gain CpGs was modestly but significantly higher in the HFD group (p=0.001; *Figure 3e*). Entropy of the age-loss and age-ns CpGs was not different between the diets. Body weight, on the other hand, was associated specifically with the entropy score of the age-loss CpGs, and both higher BW0 (*Figure 3f*) and BWF predicted higher entropy for age-loss CpGs.

We applied a multivariable regression to compare the relative effects of age, diet, BWF, glucose, cholesterol, and strain median lifespan (*Figure 3g*; full statistics in *Supplementary file 7*). Entropy of age-gain CpGs was increased by HFD but was not associated with BWF. Strain median lifespan showed a significant inverse correlation with the entropy of age-gain CpGs with an explained variance of 6% (*Figure 3h*). Entropy of the age-loss CpGs had a significant positive correlation with BWF (*Figure 3i*), but was not associated with diet, and also had a modestly significant inverse correlation with median lifespan. Cholesterol was unrelated to the entropy values. Glucose, on the other hand, showed a significant inverse association with entropy of both age-gain (*Figure 3j*) and age-loss CpGs, and this suggests slightly lower entropy with higher fasted glucose.

Taken together, our results show that the conserved CpGs are influenced by multiple predictors. HFD augmented the age-dependent changes with a prominent effect on age-gain CpGs. Body weight showed a strong association with the age-loss CpGs. Additionally, strains with longer life expectancy tended to have lower methylation levels at age-gain CpGs with an overall lower entropy state at these CpGs that suggests a more 'youthful' methylome for longer lived genotypes.

## Functional and genomic context of DMCs

To uncover the potential biological pathways represented by the DMCs, we performed genomic regions enrichment analyses for the CpGs (*McLean et al., 2010*). The age-gain CpGs were highly enriched in transcription factors, regulators of development and growth, menarche and menstrual phases, energy metabolism, and transcription factor networks such as HNF1 and HNF3B pathways (*Supplementary file 8*). The age-loss CpGs had somewhat modest enrichment and represented cell adhesion and cytoskeletal processes, endothelial cell proliferation, and p38 signaling. The BW-DMCs were enriched in actin and protein metabolism, and WNT, and platelet-derived growth factor (PDGF) and ErbB signaling. Similarly, the diet-DMCs were highly enriched in PDGF, epidermal growth factor (EGFR), and ErbB signaling, as well as the mTOR signaling pathway, and regulation of energy homeostasis (*Supplementary file 8*). Seeming to converge on common pathways, the LS-CpGs that were negatively correlated with lifespan had modest enrichment in cell signaling pathways such as EGFR, PDGF, and ErbB signaling. The LS-CpGs with positive correlation with lifespan were highly enriched in lipid metabolic genes and also included pathways related to chromosome maintenance and telomere expansion (*Supplementary file 8*).

We next examined the genomic annotations and chromatin states of the DMCs. Consistent with previous reports (*Sandoval-Sierra et al., 2020*; *Sziráki et al., 2018*), age-gain CpGs were enriched in promoter and 5'UTR CpGs, but depleted in 3'UTR, exon, and intergenic CpGs (*Figure 2—figure supplement 1a*, *Supplementary file 9*). Diet- and weight-DMCs were depleted in promoter regions and enriched in exons and 3'UTR, and along with the age-loss CpGs, enriched in introns. For chromatin states, we annotated the CpG regions using the 15-state chromatin data for neonatal (P0) mouse liver (*Ernst and Kellis, 2012*; *Gorkin et al., 2020*). Also included were regions labeled as no reproducible state (NRS); that is, regions that were not replicated (*Supplementary file 6* has annotations for each site; *Gorkin et al., 2020*). Compared to the array content as background, the age-gain CpGs were selectively enrich in polycomb-associated heterochromatin (Hc-P) and bivalent promoters (Pr-Bi), chromatin states that were highly depleted among the other DMCs (*Figure 2—figure supplement 1b*). In contrast, strong and permissive transcription sites (Tr-S and Tr-P, respectively) were depleted among the age-gain CpGs and enriched among the BW- and diet-DMCs. Age-loss CpGs were enriched in Tr-P and Tr-I (transcription initiation). Distal enhancers (strong distal or En-Sd, and poised distal or En-Pd) were also highly enriched among the BW- and diet-DMCs, and also showed some enrichment among the age-DMCs.

For an overview of the general methylation and variance patterns by chromatin annotations, we used the full set of 27,966 CpGs and computed the average methylation beta-values, and average

regression coefficients (i.e., change in beta-value per unit change in the respective predictor variable, or contrast between diets). As expected, promoter CpGs and Hc-P were sites with the lowest methylation. Hc-H, Tr-S, and Tr-P had higher methylation, and many of the enhancer sites were in the hemi-methylated zone. For age effect, mean regression estimates had a significant inverse linear fit with mean methylation (r = –0.63, p=0.009; *Figure 2—figure supplement 1c*), and this is consistent with the greater age-loss at hypermethylated CpGs and greater age-gains at hypomethylated CpGs (*Figure 2f*). The effects of diet and weight were not linearly related to the mean methylation of the chromatin states. Instead, both showed a U-shaped fit with a significant negative quadratic effect for diet ($R^2$ = 0.69, p=0.0005, quadratic estimate = –0.05; *Figure 2—figure supplement 1d*) and a positive quadratic effect for weight ($R^2$ = 0.50, p=0.01, quadratic estimate = 0.001; *Figure 2—figure supplement 1e*). Methylation variation as a function of strain longevity did not relate to mean methylation with either a linear or polynomial fit, and indicates that variance due to background genotype is less dependent on the chromatin and mean methylation status. While this is a very low-resolution and broad view of methylation levels and methylation variation, the observations show that while aging results in erosion of the hypo- and hypermethylated peaks, diet and body weight appear to have generally stronger associations with hemi-methylated sites.

## Genetic analysis of epigenetic age acceleration

The EAA traits had moderate heritability at an averaged $h^2$ of 0.50 (*Table 2*; *Ashbrook et al., 2021*). Another way to gauge the level of genetic correlation is to compare between members of strains maintained on different diets. All the EAA traits shared high strain-level correlations between diets, indicating an effect of background genotype that is robust to dietary differences (*Table 2*). The methylome-wide entropy had a heritability of ~0.30 and had no strain-level correlation between diets.

To uncover genetic loci, we applied QTL mapping using mixed linear modeling that corrects for the BXD kinship structure (*Zhou and Stephens, 2014*). First, we performed the QTL mapping for each EAA trait with adjustment for diet and body weight. EAA from the two interventional clocks had the strongest QTLs (*Supplementary file 10*). The pan-tissue int.EAA had a significant QTL on Chr11 (90–99 Mb) with the highest linkage at ~93 Mb (p=3.5E-06; equivalent to a LOD score of 4.7) (*Figure 4a*). Taking a genotype marker at the peak interval (BXD variant ID DA0014408.4 at Chr11, 92.750 Mb) (*Ashbrook et al., 2021*), we segregated the BXDs homozygous for either the D2 (*DD*) or the B6 (*BB*) alleles. Strains with *DD* genotype at this locus had significantly higher int.EAA (*Figure 4a*, inset). The liver int.EAA had the highest QTL on Chr19 (35–45 Mb) with the most significant linkage at markers between 38 and 42 Mb (p=9E-07; LOD score of 5.2) (*Figure 4b*). We selected a marker at the peak interval (rs48062674 at Chr19, 38.650 Mb), and the *BB* genotype had significantly higher liver int.EAA compared to *DD* (*Figure 4b*, inset).

We performed a similar QTL mapping for methylome-wide entropy with adjustment for major covariates (diet, chronological age, and body weight). There were no genome-wide significant QTLs. A region on Chr19 that overlapped the liver int.EAA showed a modest peak (*Figure 4c*, *Supplementary file 10*). However, the peak markers for entropy were located slightly distal to the peak EAA QTL (~47.5 Mb at rs30567369, minimum p=0.0005). At this locus, the *BB* genotype had higher average entropy.

To identify regulatory loci that are consistent across the different EAA measures, we applied a multitrait analysis and derived the linkage meta-p-value using a p-value combination for the six EAA traits (*Peirce et al., 2007*). The peaks on Chrs 11 and 19 attained the highest consensus p-values (*Figure 4—figure supplement 1a*). There was another potential consensus peak at combined $-\log_{10}$ p>6 on Chr3 (~54 Mb). We focus on the Chrs 11 and 19 QTLs and refer to these as EAA QTL on Chr 11 (Eaa11), and EAA QTL on Chr 19 (Eaa19). Eaa11 extends from 90 to 99 Mb. For Eaa19, we delineated a broader interval from 35 to 48 Mb that also encompasses the peak markers for entropy.

We performed marker-specific linkage analyses for each of the clocks using a regression model that adjusted for diet. With the exception of the liver int.EAA, all the EAA traits had nominal to highly significant associations with the representative Eaa11 marker (DA0014408.4), and the *DD* genotype had higher age acceleration (*Table 3*). Mean plots by genotype and diet show that this effect was primarily in the CD mice (*Figure 4—figure supplement 1b*). The effect of this locus appeared to be higher for the pan-tissue clocks compared to the corresponding liver-specific clocks. For proximal Eaa19, the representative marker (rs48062674) was associated with all the EAA traits and the *BB* mice

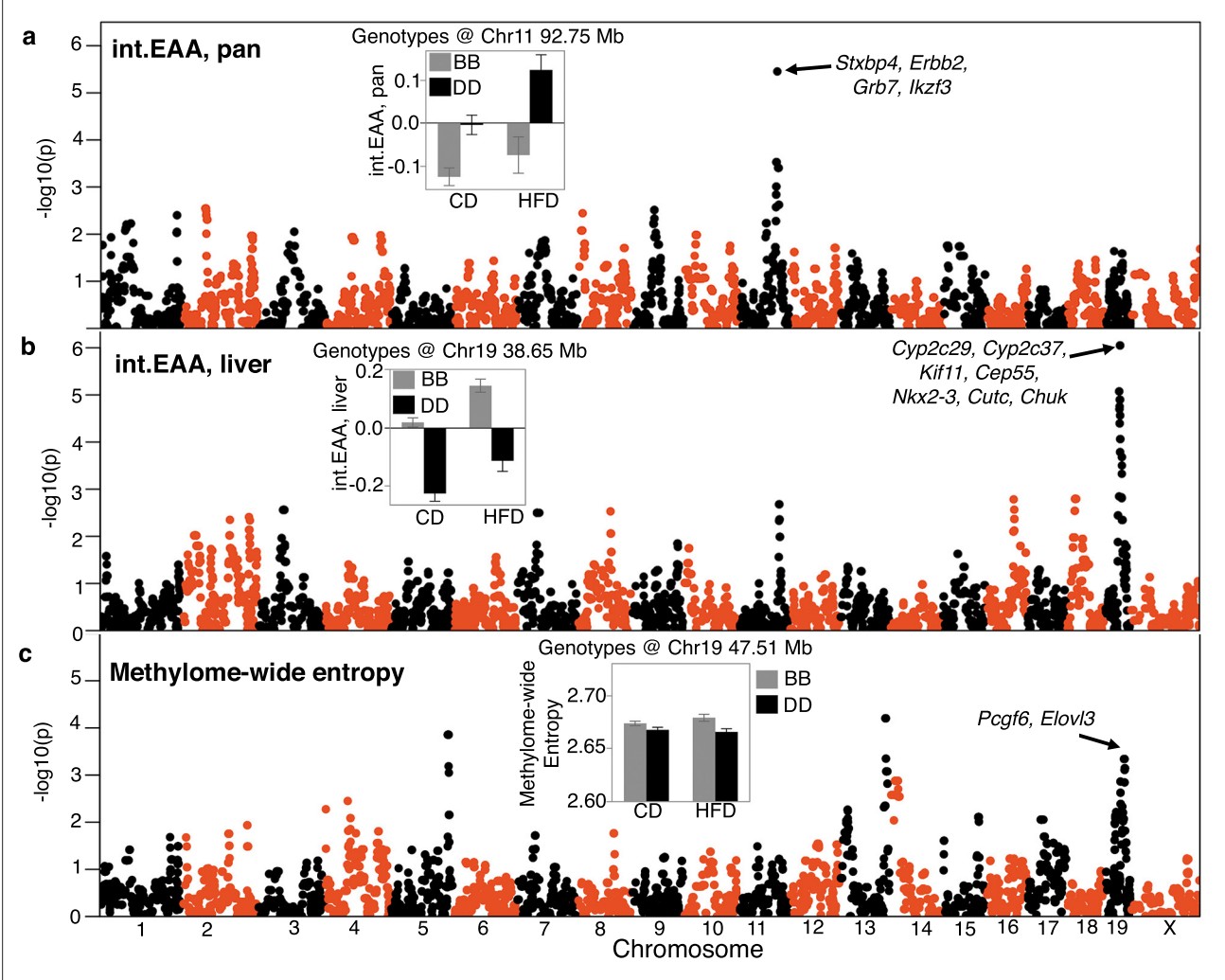

**Figure 4.** Quantitative trait locus (QTL) maps for the DNAm readouts. The Manhattan plots represent the location of genotyped markers (x-axis) and linkage $-\log_{10}p$ (y-axis). (**a**) The peak QTL for age acceleration from the pan-tissue interventional clock (int.EAA) is on chromosome (Chr) 11 at ~93 Mb. The inset shows the mean (± standard error) trait values for BXDs homozygous for the C57BL/6J allele (*BB*; gray) versus BXDs homozygous for the DBA/2J allele (*DD*; black) on control diet (CD) and high-fat diet (HFD). (**b**) The liver-specific int.EAA has a peak QTL on Chr19 (~38 Mb). Trait means by genotype at this locus are shown in inset; *BB* has higher age acceleration. (**c**) Linkage statistics are weaker for the methylome-wide entropy. However, there is a nominally significant linkage on the Chr19 locus, but the peak markers are at ~47.5 Mb. Here, the *BB* genotype has higher entropy.

The online version of this article includes the following figure supplement(s) for figure 4:

**Figure supplement 1.** Consensus quantitative trait locus (QTL) mapping for epigenetic age acceleration (EAA).

had higher age acceleration in both diets (*Figure 4—figure supplement 1c*). We also tested if these peak markers were associated with the recorded lifespan phenotype, and we found no significant association with the observed lifespan of the BXDs.

## Association of EAA QTLs with body weight trajectory

Since gain in body weight with age was an accelerator of the clocks, we examined whether the selected markers in Eaa11 and Eaa19 are also related to body weight change. We retrieved longitudinal weight data from a larger cohort of the aging BXD mice that were weighed at regular intervals. After excluding heterozygotes, we tested the effect of genotype. Concordant with the higher EAA for the *DD* genotype at Eaa11 in the CD group, the *DD* genotype in the CD group also had slightly higher mean weight in older adulthood (12 and 18 months; *Figure 5a*). However, this marker had no significant association with body weight when tested using a mixed-effects model (p=0.07; *Table 3*). In Eaa19, it was the *BB* genotype that consistently exhibited an accelerated clock in both diets, and also

**Table 3.** Marker-specific linkage analyses for epigenetic age acceleration and body weight trajectory.

| Predictor | Outcome | Estimate | Standard error | t ratio | p |
|---|---|---|---|---|---|
| | **Linear regression*** | | | | |
| Eaa11 DA0014408.4[DD] Chr11, 92.750 Mb (133 *BB* cases, and 173 *DD* cases) | EAA, pan | 0.096 | 0.023 | 4.184 | 3.8E-05 |
| | EAA, liver | 0.067 | 0.017 | 3.880 | 0.0001 |
| | dev.EAA, pan | 0.077 | 0.025 | 3.041 | 0.003 |
| | dev.EAA, liver | 0.037 | 0.020 | 1.878 | 0.06 |
| | int.EAA, pan | 0.153 | 0.029 | 5.278 | 2.5E-07 |
| | int.EAA, liver | –0.033 | 0.025 | –1.284 | 0.20 |
| Eaa19 rs48062674[DD] Chr19, 38.650 Mb (238 *BB* cases, and 67 *DD* cases) | EAA, pan | –0.083 | 0.028 | –2.954 | 0.003 |
| | EAA, liver | –0.137 | 0.020 | –6.972 | 2.0E-11 |
| | dev.EAA, pan | –0.206 | 0.029 | –7.218 | 4.3E-12 |
| | dev.EAA, liver | –0.124 | 0.023 | –5.461 | 9.9E-08 |
| | int.EAA, pan | –0.143 | 0.035 | –4.028 | 7.1E-05 |
| | int.EAA, liver | –0.250 | 0.027 | –9.238 | 4.6E-18 |
| | **Mixed model for longitudinal change in body weight†** | | | | |
| **Predictor** | **Outcome** | **Estimate** | **Standard error** | **t ratio** | **p** |
| Eaa11 DA0014408.4[DD] Number of observations = 6885; Number of individuals = 2112 | Body weight | 0.619 | 0.345 | 1.794 | 0.07 |
| Eaa19 rs48062674[DD] Number of observations = 6132; Number of individuals = 1852 | Body weight | –1.847 | 0.374 | –4.945 | 7.6E-07 |

int, interventional; dev, developmental; EAA, epigenetic age acceleration.

*Regression model: lm(EAA ~ genotype + diet).

†lmer(weight ~age + diet + genotype + (1|mouseID)).

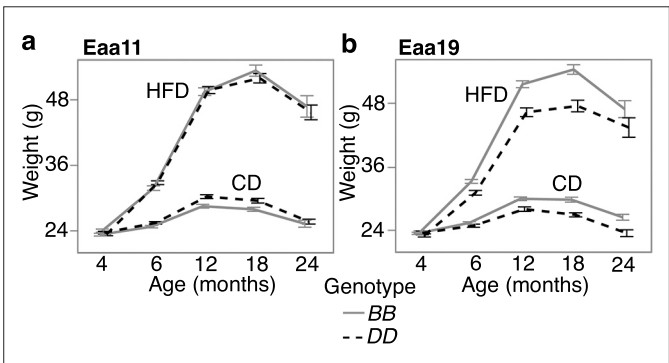

**Figure 5.** Body weight trajectory by diet and genotype. Body weight was measured at regular age intervals (x-axis) from (**a**) 2112 BXD mice that were homozygous at the Eaa11 marker (DA0014408.4; 842 *BB*, 1279 *DD*) and (**b**) 1852 BXD mice that were homozygous at the proximal Eaa19 marker (rs48062674; 1252 *BB*, 600 *DD*). Mice were maintained on either control diet (CD) or high-fat diet (HFD). The graphs show the segregation of body weight over time by diet and genotype. Mean ± standard error; heterozygotes were excluded.

higher entropy, and the *BB* genotype had higher average body weight by 6 months of age (*Figure 5b*), and this locus had a significant influence on the body weight trajectory (p=7.6E-07; *Table 3*).

## Candidate genes for epigenetic age acceleration

There are several positional candidate genes in Eaa11 and Eaa19. To narrow the list, we applied two selection criteria: genes that (1) contain missense and/or stop variants, and/or (2) contain non-coding variants and regulated by *cis*-acting expression QTLs (eQTL). For the eQTL analysis, we utilized an existing liver transcriptome data from the same aging cohort (*Williams et al., 2022*). We identified 24 positional candidates in Eaa11 that includes *Stxbp4*, *Erbb2* (*Her-2* oncogenic gene), and *Grb7* (growth factor receptor binding) (*Supplementary file 11*, *Figure 4a*). Eaa19 has 81 such candidates that includes a cluster of cytochrome P450 genes, and *Chuk* (inhibitor of NF-kB) in the proximal region, and *Pcgf6* (epigenetic regulator) and *Elovl3* (lipid metabolic gene) in the distal region (*Supplementary file 11*, *Figure 4b and c*).

For further prioritization, we converted the mouse QTL regions to the corresponding syntenic regions in the human genome and retrieved GWAS annotations for these intervals (*Buniello et al., 2019*). We specifically searched for the traits: epigenetic aging, longevity, age of menarche/menopause/puberty, Alzheimer's disease, and age-related cognitive decline and dementia. This highlighted five genes in Eaa11 and three genes in Eaa19 (*Supplementary file 4c*). We also identified a GWAS that found associations between variants near *Myof-Cyp26a1* and human longevity (*Yashin et al., 2018*), and a meta-GWAS that found gene-level associations between *Nkx2–3* and *Cutc*, and epigenetic aging (*Supplementary file 4c*; *McCartney et al., 2021*).

## Gene expression correlates of EAA

A subset of the BXD cases had liver RNA-seq data (94 CD and 59 HFD) (*Williams et al., 2022*). Using this set, we performed transcriptome-wide correlation analysis for the general pan-tissue EAA and the more specific liver int.EAA. To gain insights into biological pathways, we selected the top 2000 transcriptome correlates for functional enrichment analysis (*Supplementary file 12a*). The common themes for both clocks were that (1) there were far fewer negative correlates (223 out of 2000 for pan-tissue EAA, and 337 out of 2000 transcripts for liver int.EAA) than positive correlates, and (2) the negative correlates were highly enriched (Bonferroni corrected p<0.05) in oxidation-reduction and mitochondrial genes (*Supplementary file 12b and c*). The pan-tissue general clock was also highly enriched in pathways related to steroid metabolism, epoxygenase p450 pathway, and xenobiotics, which are pathways that are particularly relevant to liver. The p450 genes included candidates that are in Eaa19 (e.g., *Cyp2c29*, *Cyp2c37*). The positive correlates were enriched in a variety of gene functions, including mitosis for both clocks, and immune and inflammatory response for the general pan-tissue clock (functions that are not specific to liver). 563 transcripts (315 unique genes) were correlated with both pan-tissue EAA and liver int.EAA. Based on hierarchical clustering (HC) of these common mRNA correlates of EAA, the transcripts could be clustered into three groups (*Figure 6a*; heatmap in *Figure 6—figure supplement 1a*). While none of these were significantly enriched in any particular Gene Ontology (GO), cluster 3 included several oxidation-reduction genes, including the Eaa11 candidate, *Cyp2c29*, and cluster 2 included several cell cycle genes (*Figure 6a*). To verify that these transcriptomic associations are robust to the effect of diet, we repeated the correlation and enrichment analysis in the CD group only for the pan-tissue general clock (n = 94). Again, taking the top 2000 correlates (|r| ≤ 0.22; p≤0.03), we found the same enrichment profiles for the positive correlates (immune, cell cycle) and the negative correlates (oxidation-reduction and mitochondrial).

Liver proteome was also available for 164 of the BXDs, and 53 also had adipose proteome. The liver proteome data quantifies over 32,000 protein variants from 3940 unique genes and has been reported in *Williams et al., 2022*. Similar to the transcriptome-wide analysis, we extracted the top 2000 protein correlates of EAA (*Supplementary file 12d*) and performed functional enrichment analysis (*Supplementary file 12b and c*). For both liver int.EAA and pan-tissue EAA, the top liver protein correlate was APOE, and higher expression of APOE was associated with higher age acceleration (*Figure 6b and c*). Similar to the transcriptome, the negative correlates of EAA were highly enriched in oxidation-reduction (several cytochrome proteins), steroid metabolism, and epoxygenase 450 pathway. The positive correlates were also highly enriched in oxidation-reduction (several hydroxy-delta-5 steroid dehydrogenases proteins), lipid and carbohydrate metabolism, as well as phospholipid

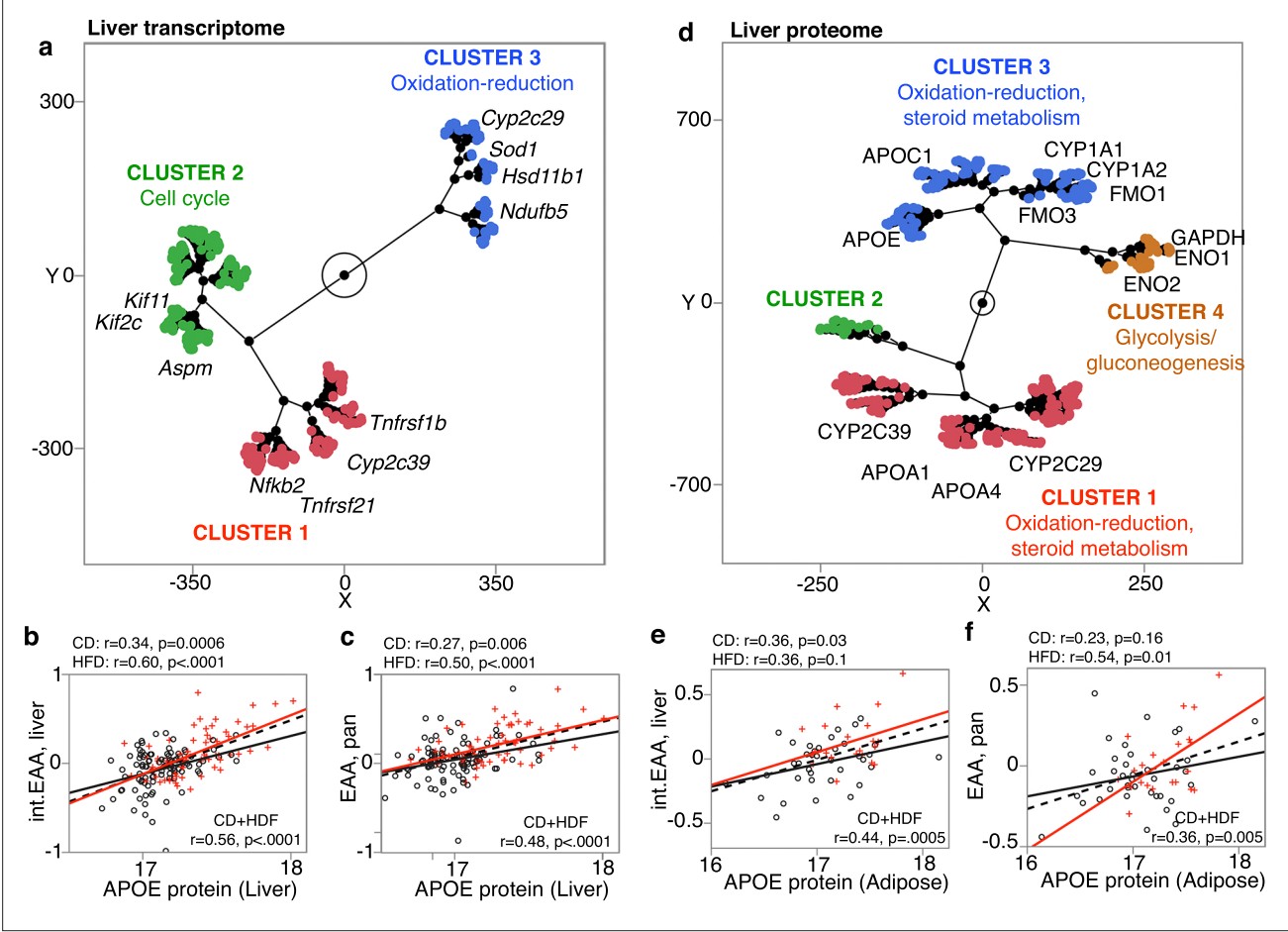

**Figure 6.** Gene expression correlates of epigenetic age acceleration (EAA). (**a**) mRNAs that are correlated with the acceleration of both pan-tissue general clock (pan EAA) and liver interventional clock (liver int.EAA) were grouped based on unsupervised hierarchical clustering (HC). Few representative genes and gene ontologies are highlighted. For liver proteome, the level of APOE is the strongest correlate for both (**b**) liver int.EAA and (**c**) pan-tissue EAA. (**d**) For liver proteins that are correlated with both pan-tissue EAA and liver int.EAA, HC grouped the proteins into clusters enriched in oxidation-reduction and lipid metabolism, and a cluster enriched in glycogen metabolism. In adipose tissue, the expression level of the APOE protein is higher with higher age acceleration for both (**e**) liver int.EAA and (**f**) pan-tissue EAA.

The online version of this article includes the following figure supplement(s) for figure 6:

**Figure supplement 1.** Hierarchical clustering heatmaps for the top expression correlates of epigenetic age acceleration.

efflux (particularly enriched for the liver int.EAA). There was a high degree of overlap at the proteomic level for the two clocks, and 1241 protein variants (332 unique genes) were correlated with both pan-tissue EAA and liver int.EAA (***Supplementary file 12d***). For these common protein correlates, the HC divided the proteins into clusters that represented metabolic pathways mainly related to steroid metabolism, and also glycolysis and gluconeogenesis (***Figure 6d***; heatmap in ***Figure 6—figure supplement 1b***).

Finally, we used the adipose proteome data for a proteome-wide correlational analysis for the pan-tissue EAA and liver int.EAA. We took only the top 1000 correlates (due to the small sample size), and a functional enrichment analysis showed consistent enrichment in metabolic pathways related to fatty acids and carbohydrates, and cell proliferation genes for the pan-tissue EAA (***Supplementary file 12b and c***). For the adipose proteome, the cytochrome p450 genes were no longer enriched. However, the overall functional profile highlighted metabolic pathways as important gene expression correlates of EAA. Furthermore, for both the liver and adipose proteomes, APOE levels were highly correlated with EAA that indicates a higher level of this apolipoprotein in both tissues is associated with higher age acceleration (***Figure 6e and f***).

## Discussion

Here, we have tested the performance of DNAm clocks derived from highly conserved CpGs and described the dynamism and variability of site-specific methylation. While age is a major source of variance, we detected joint modulation by diet, body weight, and genotype-by-diet life expectancy. HFD had an age-accelerating effect on the clocks, and this is concordant with our previous report where we found more rapid age-dependent changes in methylation (*Sandoval-Sierra et al., 2020*). This also concurs with studies in humans that have found that obesity accelerates epigenetic aging (*Horvath et al., 2014*; *Nevalainen et al., 2017*). However, when BWF was included in the regression term, the effect of diet became inconsistent. This suggests that the effect of diet on EAA is mediated by the changes in weight and metabolic traits. Body weight, in particular, had a strong age-accelerating effect. The effect of weight manifests early on, and even in the CD group, higher weight gains at younger age (between 4 and 6 months) correlated with higher EAA later in life.

We tested different mouse DNAm clocks, and the main difference between these clocks was the subsets of CpGs that were used for training. It is well-known that DNAm clocks have high level of degeneracy (*Thompson et al., 2018*; *Liu et al., 2020*). In other words, highly accurate predictors of chronological age can be built from entirely different sets of CpGs and different weight coefficients. This is likely because a large proportion of CpGs undergo some degree of change with age, and combinatorial information from any subset of such CpGs is informative of age. For instance, even at a very stringent cutoff of Bonferroni 0.05 that treated the 27,966 CpGs as 'independent,' we still detected 6553 CpGs as age-DMC, which is close to a quarter of the CpGs we tested. Clocks built from pre-selected CpGs that are at conserved sequences are known to be sensitive to the effects of pro-longevity interventions such as caloric restriction and growth hormone receptor deletion (*Thompson et al., 2018*; *Wang and Lemos, 2019*). And while all these DNAm clocks achieve reasonably high prediction of chronological age, the age divergence derived from these different clocks (EAA) can capture slightly different facets of biological aging, and the better a clock is at predicting chronological age, the lower its association with mortality risk (*Zhang et al., 2019*; *Liu et al., 2020*). In this study, we find that the interventional clocks deviated most from chronological age, and this is expected as these were built from a much smaller set of CpGs (see Materials and methods). The interventional clocks were also associated with BWF and cholesterol, but had weaker associations with BW0. The liver int.EAA had the highest positive correlation with methylome-wide entropy and was the clock that had the strongest inverse correlation with strain longevity. In contrast, the developmental clocks, which were based on CpGs that change early in life, showed a stronger association with BW0. The contrast between the interventional and developmental clocks suggests that while one is more modifiable, the other is more informative of baseline characteristics that influence aging later in life. The pan-tissue general clock, which was not constrained to any preselected set of CpGs or tissue, also performed well in capturing biological aging and was accelerated by both BW0 and BWF, diet, higher entropy, and had a modest but significant inverse correlation with strain lifespan.

Entropy, a measure of noise and information loss, increases as a function of time and age (*Hannum et al., 2013*; *Xie et al., 2011*; *Jenkinson et al., 2017*; *Hayflick, 2007*). In the context of the methylome, the shift to higher entropy represents a tendency for the highly organized hypo- and hyper-methylated landscape to erode towards a more hemi-methylated state (*Hannum et al., 2013*; *Sziráki et al., 2018*; *Kerepesi et al., 2022*). This increase in disorder, particularly across CpGs that are highly conserved, could have important functional consequences. The entropy of age-gain CpGs predicted strain lifespan and was increased by HFD. Overall, we find that mice belonging to longer-lived BXD strains had a more 'youthful' methylome with lower entropy at the age-gain CpGs. The entropy of age-loss CpGs, on the other hand, was related to the body weight of mice, and both higher BW0 and BWF were associated with higher entropy. This leads us to suggest that the rate of noise accumulation, an aspect of epigenomic aging, can vary between individuals, and the resilience or susceptibility to this shift towards higher noise is modulated by diet as well as genetic factors.

Somewhat surprising was the inverse correlation between the entropy of age-DMCs and fasted glucose. This lower entropy of age-gain CpGs with higher glucose is counter to the general tendency for strains with shorter lifespan to have higher glucose (*Roy et al., 2021*). In biological systems, entropy is kept at bay by the uptake of chemical energy and investment in maintenance and repair (*Hayflick, 2007*), and we can only speculate that at least in mice, the higher amount of glucose after overnight fast is associated with a more ordered methylome. The centrality of bioenergetics for

biological systems may explain why we detect this coupling between the DNAm readouts (i.e., the clocks and entropy), and indices of metabolism including weight, diet, levels of macronutrients, and even expression of metabolic genes. As cogently highlighted by *Donohoe and Bultman, 2012*, many metabolites (e.g., SAM, NAD⁺, ATP) are essential cofactors for enzymes that shape the epigenome, and these could serve as nutrient sensors and mechanistic intermediaries that regulate how the epigenome is organized in response to metabolic conditions. Close interactions between macro- and micronutrients, and DNAm is a conserved process and plays a critical role in defining both physiology and body morphology (*Kucharski et al., 2008*; *Dolinoy et al., 2007*). Overall, our results suggest that a higher metabolic state is associated with higher entropy and EAA, and potentially, lower lifespan.

For the BXDs, life expectancy is highly dependent on the background genotype, and mean lifespan varies from under 16 months for strains such as BXD8 and BXD13, to over 28 months in strains such as BXD91 and BXD175 (*Roy et al., 2021*; *de Haan and Williams, 2005*; *Lang et al., 2010*). The EAA showed the expected inverse correlation with lifespan, but the effect was modest and only significant for the pan-tissue EAA and the liver int.EAA. The correlation between lifespan and entropy of age-gain CpGs was slightly stronger. We must point out that the analysis between the epigenetic readouts and lifespan was an indirect comparison. Unlike the comparison with body weight and metabolic traits, which were traits measured from the same individual, the lifespan data are strain characteristics computed from a parallel cohort of mice that were allowed to survive till natural mortality, and this may partly explain the weaker associations with EAA. Nonetheless, our observations indicate that genotypes with higher life expectancy have generally lower entropy, and lower methylation levels at the age-gain CpGs, and these properties of the methylome are likely to be partly under genetic modulation.

Our goal was to take these different clocks and identify regulatory loci that were the most stable and robust to the slight algorithmic differences in building the clocks. A notable candidate in Eaa11 is Syntaxin binding protein 4 (*Stxbp4*, aka, *Synip*), located at 90.5 Mb. *Stxbp4* is a high-priority candidate due to the concordant evidence from human genetic studies. The conserved gene in humans is a replicated GWAS hit for the intrinsic rate of epigenetic aging (*Gibson, 2019*; *Lu et al., 2018*; *McCartney et al., 2021*). In the BXDs, *Stxbp4* contains several noncoding variants, and a missense mutation (rs3668623), and the expression of *Stxbp4* in liver is modulated by a *cis*-eQTL. *Stxbp4* plays a key role in insulin signaling (*Holman, 1999*) and has oncogenic activity and implicated in different cancers (*Michailidou et al., 2017*; *Rokudai et al., 2018*). Furthermore, GWAS have also associated *STXBP4* with age of menarche (*Kichaev et al., 2019*; *Perry et al., 2014*). Eaa11 corresponds to the 17q12-21 region in humans, and the location of additional oncogenic genes, for example, *ERBB2/HER2*, *GRB7*, and *BRCA1* (*Albertsen et al., 1994*). The mouse *Brca1* gene is a little distal to the peak QTL region and is not considered a candidate here, although it does segregate for two missense variants in the BXDs. *Erbb2* and *Grb7* are in the QTL region, and *Erbb2* contains a missense variant (rs29390172), and *Grb7* is modulated by a *cis*-eQTL. *Nr1d1* is another candidate in Eaa11, and the co-activation of *Erbb1*, *Grb7*, and *Nr1d1* has been linked to breast and other cancers (*Kauraniemi and Kallioniemi, 2006*; *Tanaka et al., 1997*).

Eaa19 was consistently associated with EAA from all the clocks we evaluated, and also with body weight gains, irrespective of diet. DNAm entropy may also have a weak association with markers at this interval. The EAA traits have peak markers in the proximal part of Eaa19 (around the cytochrome cluster), and the methylome-wide entropy had a weak peak that was in the distal portion (over candidates like *Elovl3*, *Pcgf3*). Two candidates in Eaa19 have been implicated in epigenetic aging in humans based on gene-level meta-GWAS: NK homeobox 3 (*Nkx2-3*, a developmental gene) and CutC copper transporter (*Cutc*) (*McCartney et al., 2021*). Eaa19 is also the location of the *Cyp26a1-Myof* genes, and the human syntenic region is associated with longevity, metabolic traits, and lipid profiles (*Yashin et al., 2018*; *de Vries et al., 2019*; *Richardson et al., 2020*). Another noteworthy candidate in Eaa19 is *Chuk*, a regulator of mTORC2, that has been associated with age at menopause (*Kichaev et al., 2019*; *Xu et al., 2013*). Eaa19 presents a complex and intriguing QTL related to the DNAm readouts that may also influence body weight gains over the course of life. Both Eaa19 and Eaa11 exemplify the major challenge that follows when a genetic mapping approach leads to gene- and variant-dense regions (*Gallagher and Chen-Plotkin, 2018*; *Lappalainen, 2015*). Both loci have several biologically relevant genes, and identifying the causal gene (or genes) will require a more fine-scaled functional genomic dissection.

The gene expression analyses highlighted metabolic pathways. At the mRNA level, the negative correlates of EAA were highly enriched in metabolic genes related to oxidation-reduction and steroid metabolism, while the positive correlates were enriched in pathways related to mitosis, and immune response for the pan-tissue general EAA. This convergence of metabolic, immune, and cell division genes is very consistent with previous reports (*Liu et al., 2020*; *Kuo et al., 2021*; *Slieker et al., 2016*). Here, we should note that depending on the tissue(s) in which the clocks are trains, and the tissue from which the DNAmAge is estimated, the EAA derivative may put an emphasis on biological pathways or genes that are most relevant to that tissue. For instance, clocks optimized for neural tissue are more closely related to neurodegeneration and neuropathologies (*Shireby, 2020*; *Grodstein et al., 2021*). With the liver clocks, expression correlates highlighted aspects of metabolism that are relevant to liver function (e.g., the cytochrome p450 epoxygenase genes), and this is detected both at the transcriptomic and proteomic levels. For the adipose tissue proteome, the cytochrome genes become less prominent, but the enriched pathways still remained consistent (i.e., oxidation-reduction, lipid and carbohydrate metabolism, and cell proliferation for the positive correlates of the pan-tissue EAA). At the proteome level, we also find several phospholipid efflux genes (APOC1, APOA2, APOC3, APOA1, APOA4, APOE) that are positive correlates of EAA. For both the liver and adipose proteomes, APOE stands out as the top protein correlate of EAA. A recent human study has also identified the APOE locus as the strongest GWAS hit for two measures of biological age acceleration (the phenoAge and the bioAge) (*Kuo et al., 2021*). While more specific to liver, the cytochrome P450 genes presents as both positional candidates and expression correlates of EAA. These genes have high expression in liver and have major downstream impact on metabolism (*Olona et al., 2018*; *Schuck et al., 2014*; *Wang et al., 2020*). One caveat is that these CYP genes are part of a gene cluster in Eaa19 that includes transcripts with *cis*-eQTLs (e.g., *Cyp2c66*, *Cyp2c39*, *Cyp2c68*) and the tight clustering of the genes, and proximity of trait QTL and eQTLs may result in co-expression due to linkage disequilibrium (*Mozhui et al., 2008*). Nonetheless, the cytochrome genes in Eaa19 are strong candidate modulators of EAA derived from liver tissue that calls for further investigation.

Aside from Eaa11 and Eaa19, another locus with evidence for consensus QTL was detected on Chr3. We do not delve much into this here, but the Chr3 interval is near genes associated with human epigenetic aging (*Ift80*, *Trim59*, *Kpna4*) (*Gibson, 2019*; *McCartney et al., 2021*). However, this QTL is dispersed across a large interval, and the peak markers do not exactly overlap these human EAA GWAS hits. While we have focused on Eaa11 and Eaa19, the Chr3 locus presents a potentially important region for EAA.

In summary, we have identified two main QTLs – Eaa11 and Eaa19 – that contribute to variation in EAA. Eaa11 contains several genes with oncogenic properties (e.g., *Stxbp4*, *Erbb2*), while Eaa19 contains a dense cluster of metabolic genes (e.g., *Elovl3*, *Chuk,* the cytochrome genes). We demonstrate that metabolic profile and body weight are closely related to epigenetic aging and methylome entropy. The convergence of evidence from genetic and gene expression analyses indicates that genes involved in metabolism and energy balance contribute to the age-dependent restructuring of the methylome, which in turn forms the basis of the epigenetic clocks.

# Materials and methods
## Biospecimen collection and processing

Samples for this study were selected from a larger colony of BXD mice that were housed in a specific pathogen-free facility at the University of Tennessee Health Science Center (UTHSC). All animal procedures were in accordance with a protocol approved by the Institutional Animal Care and Use Committee (IACUC) at the UTHSC. Detailed description of housing conditions and diet can be found in *Williams et al., 2022*; *Roy et al., 2021*. Mice were given ad libitum access to water and either standard laboratory chow (Harlan Teklad; 2018, 18.6% protein, 6.2% fat, 75.2% carbohydrates) or high-fat chow (Harlan Teklad 06414; 18.4% protein, 60.3% fat, 21.3% carbohydrate). Animals were first weighed within the first few days of assignment to either diets, and this was mostly but not always prior to introduction to HFD. Following this, animals were weighed periodically and a final time (BWF) when animals were humanely euthanized (anesthetized with avertin at 0.02 ml/g of weight, followed by perfusion with phosphate-buffered saline) at specific ages for tissue collection. This work utilizes the biobanked liver specimens that were pulverized and stored at –80°C, and overlaps samples described

in *Williams et al., 2022*. DNA was extracted using the DNeasy Blood & Tissue Kit from QIAGEN. Nucleic acid purity was inspected with a NanoDrop spectrophotometer and quantified using a Qubit fluorometer dsDNA BR Assay.

## Methylation array and quality checks

DNA samples from ~350 BXD mice were profiled on the Illumina HorvathHumanMethylChip40 array. Samples were in 96-well plate format (*Supplementary file 1*), and the plates were randomized for major covariates such as age and diet. Details of this array are described in *Arneson et al., 2022*; *Horvath and Haghani, 2021*. The array contains probes that target ~36K CpGs that are highly conserved in mammals. Over 33K probes map to homologous regions in the mouse genome. For downstream statistical tests, we further filtered the probes and used only 27,966 probes that have been validated for the mouse genome using calibration data generated from synthetic mouse DNA (*Arneson et al., 2022*). Data was normalized using the SeSame method (*Zhou et al., 2018*). Unsupervised HC was performed to identify outliers and failed arrays, and those were excluded. We also performed strain verification as an additional quality check. While the majority of the probes were free of DNA sequence variants, we found 45 probes that overlapped variants in the BXD family. We leveraged these as proxies for genotypes and performed a principal component analysis (PCA). The top genotype principal components (genoPC1 and genoPC2; *Supplementary file 1*) segregated the samples by strain identity, and samples that did not cluster close to the reported strains were removed. After excluding outliers, failed arrays, and samples that failed strain verification, the final liver DNAm data consisted of 339 samples. The beta-values for these ~28K probes in the 339 samples show the expected bimodal distribution (*Figure 2—figure supplement 2a*), but for these highly conserved CpGs, we note a much higher representation of hypermethylated CpGs instead of the slightly hypomethylated state of the methylome when a wider spectrum of CpGs is assayed (*Sziráki et al., 2018*).

## BXD-unbiased mouse clock estimation

Three different mouse clocks are reported here, and all three are based on penalized regression modeling using glmnet (*Friedman et al., 2010*). Training was done in a larger mouse dataset that excluded the BXDs (*Lu et al., 2021*; *Li et al., 2021*; *Haghani et al., 2022*). The clocks are therefore unbiased to the characteristics of the BXDs. For pan-tissue clocks, all mouse samples were used for training. For the liver-specific clocks, the training was limited to data from liver samples.

The general DNAmAge clock did not preselect for any CpGs, and the full set of CpGs that map to *Mus musculus* was used. First, a log-linear transformation was applied to the chronological age using the function:

$$f(Age) = \begin{cases} \frac{Age}{1.2+0.06} + \log\left(1.2 + 0.06\right) - \frac{1.2}{1.2+0.06}, Age > 1.2 \\ \log\left(Age + 0.06\right), \, Age \leq 1.2 \end{cases}$$

This is similar to the age transformation described in the original Horvath pan-tissue human clock, but with offset at 0.06 and adult mouse age at 1.2 (*Horvath, 2013*). Following this transformation, an elastic net regression was implemented to regress the transformed chronological age on the CpG beta-values in the training data. The alpha was set at 0.5, and the optimal lamda parameter was determined by 10-fold cross-validation (function cv.glmnet). This selected subsets of clock CpGs and coefficients. DNAmAge was then calculated as:

$$DNAmAge = f^{-1}\left(\frac{b_0 + b_1 CpG_1 + b_2 CpG_2 + \ldots + b_i CpG_i}{b_0 + b_1 + b_2 + \ldots + b_i}\right)$$

where $b_0$ is the intercept, $b_1$ to $b_i$ are the coefficients, $CpG_1$ to $CpG_i$ denote the beta-values for the respective clock CpGs, and $f^{-1}()$ denotes the inverse function of $f()$.

A similar method was used to build the developmental and interventional clocks, but for these, the CpGs were pre-selected. For the liver-specific developmental clock, CpGs that change during mouse development were selected in liver samples based on Pearson correlation with age in mice that were <1.6 months old. The top 1000 negative and top 1000 positive correlates were then classified as 'developmental CpGs', and the training was done using only this subset of CpGs. For the pan-tissue

dev.DNAmAge, the top 1000 positive and top 1000 negative developmental CpGs were based on a multitissue EWAS, also using Pearson correlation with age for mice <1.6 months old, and these are CpGs that are strongly correlated with age during the mouse developmental period when all available tissues are considered.

Training for the interventional clock started with 537 CpGs that relate to gold-standard anti-aging interventions (caloric restriction, growth hormone receptor knockout) (*Haghani et al., 2022*; *Coschigano et al., 2003*). These 'interventional CpGs' were identified from an independent mouse liver caloric restriction (n = 95) and one growth hormone receptor knockout (n = 71) data that were not included in the clock estimation (*Haghani et al., 2022*). Top CpGs associated with these interventions were identified, and the 537 CpGs are the sites that are consistently associated with these anti-aging interventions. Of the 537, 121 CpGs increased in methylation and 416 decreased in methylation with application of the pro-longevity interventions. Given the small number of CpGs that went into training for the int.DNAmAge, we expected this clock to be less correlated with chronological age, and possibly more responsive to variables such as diet.

## Entropy calculation

Methylome-wide entropy was calculated from the 27,966 probes. The beta-values were discretized into 20 bins, and the Shannon entropy for each sample was estimated using the R package, 'entropy' (v1.2.1) with method = 'ML': maximum likelihood (*Hausser and Strimmer, 2009*). The optimal number of bins was determined using the Freedman–Diaconis rule (breaks = 'FD' for the hist() function in R). We also estimated the methylome-wide entropy after discretizing into 100 and 2000 bins (values provided in *Supplementary file 1*), and the results we report are consistent and robust to the number of bins. For the age-gain, age-loss, and age-ns CpGs, entropy for each set was estimated, also following discretization into 20 bins.

## Statistics

Statistical analyses were done using R or the JMP Pro software (version 15). Associations between the epigenetic predictors and continuous variables (body weight, strain lifespan, fasted serum glucose, and total cholesterol) were based on Pearson correlations, and *t*-test was used to evaluate the effect of categorical predictors (sex, diet). Multivariable regression models were also used to control for covariates (R regression equations provided with relevant tables and supplementary files). All these traits are directly accessible from GeneNetwork 2 (GN2; more information on how to retrieve these data from GN2 is provided in *Supplementary file 13*; *Williams, 2022*; *Mulligan et al., 2017*). Longevity data was obtained from a parallel cohort of BXD mice housed in the same UTHSC colony, and members of this 'longevity cohort' were allowed to age until natural death (more detail on the longevity cohort can be found in *Roy et al., 2021*). Males were excluded and strain-by-diet lifespan summary statistics were derived. Only strain-by-diet groups with five or more observations for lifespan were included in the correlational analyses with the epigenetic predictors.

## Multivariable EWAS

Site-by-site differential methylation analysis (EWAS) was performed on the 27,966 CpGs using a multivariable regression model. As such genome-wide explorations are vulnerable to unmeasured confounders, we included the top PC derived from a PCA of the 27,966 probes (*McClay et al., 2014*). The top 10 PCs cumulatively accounted for ~62% of the variance (*Figure 2—figure supplement 2b*). A plot of PC1 (19% of variance) and PC2 (14% of variance) showed that PC1 captured some noise due to batch (*Figure 2—figure supplement 2b*). The remaining top PCs (PC2 onwards) were strongly associated with biological variables, particular age, and also weight and diet (top 10 PCs provided in *Supplementary file 1*). For this reason, we included PC1 as a correction factor in the EWAS. The regression model we used was lm(CpG$_i$ ~ age + median lifespan + diet + BWF + PC1), where CpG$_i$ is the ith CpG from 1 to 27,966. As lifespan was from female mice, this EWAS excluded the few male samples.

## CpG annotation and enrichment

Functional annotation and enrichment analyses for the DMCs were done using the genomic region enrichment R package, rGREAT (version 3.0.0; *McLean et al., 2010*) with the array content (i.e., the

27,966 CpGs) as background. Enrichment p-values are based on hypergeometric tests, and categories with Benjamini–Hochberg adjusted p-values≤0.05 are reported. Annotations were for the GRCm38/mm10 reference genome.

For chromatin state annotation, we used bedtools to annotate the 27,966 CpGs coordinates using chromatin annotation.bed files for neonatal (P0) mouse liver tissue created by *Gorkin et al., 2020*; *Quinlan and Hall, 2010*. This provides the 15-state model using ChromHMM (*Ernst and Kellis, 2012*), and we downloaded the file for the 'replicated set' (here, the regions annotated as NRS are sites that did not produce replicable signal). Enrichment and depletion analyses for genomic annotations and chromatin annotations were based on the hypergeometric test (phyper R function). The R codes are provided with the results data (*Supplementary file 9*).

## Genetic analyses

Heritability within diet was estimated as the fraction of variability that was explained by background genotype (*Ashbrook et al., 2021*; *Ashbrook et al., 2018*; *Belknap, 1998*). For this, we applied an ANOVA: aov(EAA ~ strain), and heritability was computed as $h^2 = SSq_{strain}/(SSq_{strain} + SSq_{residual})$, where $SSq_{strain}$ is the strain sum of squares, and $SSq_{residual}$ is the residual sum of squares.

All QTL mapping was done on the GN2 platform (trait accession IDs provided in *Supplementary file 13*; *Williams, 2022*). On the GN2 home page, the present set of BXD mice belongs to the Group: BXD NIA Longevity Study, and GN2 provides a direct interface to the genotype data. All QTL mapping was done for genotypes with minor allele frequency ≥ 0.05 using the genome-wide efficient mixed model association (GEMMA) algorithm (*Zhou and Stephens, 2014*), which corrects for the BXD kinship matrix. For the EAA traits, diet, weight at 6 months, and final weight were fitted as cofactor. Chronological age had no correlation with EAA, and this was not included as a cofactor (including age does not change the results). Genome-wide linkage statistics were downloaded for the full set of markers that were available from GN2 (7320 markers in *Supplementary file 10*). For the combined p-values, QTL mapping was done separately using GEMMA for each EAA traits, then the Fisher's p-value combination was applied to get the meta-p-value (*Peirce et al., 2007*). We used this method to simply highlight loci that had consistent linkage across the different EAA measures. QTL mapping for methylome-wide entropy was done using GEMMA with adjustment for chronological age, diet, weight at 6 months, and final weight.

For marker-specific linkage, we selected SNPs located at the peak QTL regions (DA0014408, rs48062674) and grouped the BXDs by their genotypes (F1 hybrids and other heterozygotes were excluded for this), and marker-specific linkage was tested using ANOVA and linear regression (R regression equation given in *Table 3*). rs48062674 is a reference variant that is already catalogued in dbSNP (*Sherry et al., 2001*) and is used as a marker in the QTL mapping. DA0014408.4 is an updated variant at a recombinant region in the Chr11 interval and within the peak QTL interval (*Ashbrook et al., 2021*). Genotypes at these markers for individual BXD samples are given in *Supplementary file 1*.

To test the effect of genotype on body weight change, body weight data measured at approximately 4 (baseline), 6, 12, 18, and 24 months were downloaded from GN2 (*Supplementary file 13*). Detailed description of these weight data is given in *Roy et al., 2021*. We then applied a mixed-effects regression model using the lme4 R package (*Bates et al., 2021*): lmer(weight ~ age + diet + genotype + (1|ID)), where ID is the identifier for individual mouse.

## Bioinformatic tools for candidate genes selection

Sequence variation between B6 and D2 in the QTL intervals (Chr11:90–99 Mb and Chr19:35–48 Mb) was retrieved from the Wellcome Sanger Institute Mouse Genomes Project database (release 1505 for GRCm38/mm10) (*Wellcome Sanger Institute Mouse Genome Project, 2022*; *Keane et al., 2011*; *Yalcin et al., 2011*). Positional candidates were required to contain at least one coding variant (missense and/or nonsense variants) or have noncoding variants with evidence of *cis*-regulation in liver tissue of the BXDs. *cis*-eQTLs for the candidate genes were obtained from the liver RNA-seq data described in *Williams et al., 2022*. An interface to search and analyze this transcriptome data is available from GN2 and is catalogued under *Group: BXD NIA Longevity Study; Type: Liver mRNA*; and *Dataset: UTHSC BXD Liver RNA-seq (Oct 19) TMP Log2*.

For human GWAS annotations, we navigated to the corresponding syntenic regions on the human genome by using the coordinate conversion tool in the UCSC Genome Browser. The Chr11 90–95 Mb interval on the mouse reference genome (GRCm38/mm10) corresponds to human Chr17:50.14–55.75 Mb (GRCh38/hg38) (40.7% of bases; 100% span). The Chr11 95–99 Mb interval in the mouse corresponds to human Chr17:47.49–50.14 Mb (29.3% of bases, 57.9% span) and Chr17:38.19–40.39 Mb (20.7% of bases, 44.1% span). Likewise, for the Chr19 QTL, the mm10 35–40 Mb corresponds to hg38 Chr10:89.80–95.06 Mb (32.2% of bases, 89.2% span), 40–45 Mb corresponds to hg38 Chr10:95.23–100.98 Mb (46.6% of bases, 95.6% span), and 45–48 Mb corresponds to hg38 Chr10:100.98–104.41 Mb (46.5% of bases, 100% span). We then downloaded the GWAS data for these regions from the NHGRI-EBI GWAS catalog (*Buniello et al., 2019*) and retained the GWAS hits that were related to aging.

## Transcriptome and proteome analyses

The liver RNA-seq data mentioned above was also used for the transcriptome-wide correlational analysis for EAA in the 153 cases that had both DNAm and RNA-seq data. We considered the top 2000 highest mRNA correlates (|r| = 0.24, p=0.003 for the pan-tissue EAA; |r| = 0.3, p=0.0002 for the liver int.EAA), and the list of transcripts was collapsed to a nonredundant list of gene symbols, which was uploaded to the DAVID Bioinformatics Database (version 2021 update) for GO enrichment analysis (*Huang et al., 2009a*; *Huang et al., 2009b*). Proteome correlational analysis was carried out using the data: *Group: BXD NIA Longevity Study; Type: Liver Proteome*; and *Dataset: EPFL/ETHZ BXD Liver Proteome CD-HFD (Nov19)*. Detailed description of this data is given in *Williams et al., 2022*. 164 BXD cases had both DNAm and liver proteomics, and similar to the RNA-seq, we selected the top 2000 correlates (|r| = 0.24, p=0.002 for both the pan-tissue EAA and liver int.EAA) for enrichment analysis.

59 of the BXD cases also have proteome data from adipose tissue (*Group: BXD NIA Longevity Study; Type: Adipose Proteome*; and *Dataset: Riken-Wu BXD Liver Proteome CD-HFD (Sep20)*). While small in sample number, we used this data to test whether we could recapitulate the same functional enrichment profiles in a different tissue. Details of sample preparation and processing steps for the adipose proteome are provided in the dataset's 'Info' page on GN2. In brief, protein was extracted from the adipose samples by first lysis in a buffer with protease inhibitor, followed by homogenization with a glass dounce and sonication. The protein fraction was isolated from the homogenate by centrifugation and processed for assay on a liquid chromatography tandem mass spectrometry (LC-M/MS) using a modified Phase Transfer Surfactant Method as described in *Masuda et al., 2008* and *Mostafa et al., 2020*. Samples were measured using a Q Exactive Plus Orbitrap LC-MS/MS System (Thermo Fisher). For each sample, 600 ng was injected and the samples were measured with data-independent acquisition (DIA). A portion of the peptides from the samples was pooled and fractionated using a Pierce High pH Reversed-Phase (HPRP) Peptide Fractionation Kit (Thermo Fisher Scientific) to generate a spectral library. For the HPRP fractions, 450 ng was injected and the samples were measured with data-dependent acquisition (DDA). For protein identification, the raw measurement files were searched against a mouse database using the (uniprot-reviewed_Mus_musculus_10090_.fasta) with Proteome Discoverer v2.4 software (Thermo Fisher Scientific). Filtered output was used to generate a sample-specific spectral library using the Spectronaut software (Biognosys, Switzerland). Raw files from DIA measurements were used for quantitative data extraction with the generated spectral library, as previously described (*Mostafa et al., 2020*). The false discovery rate was estimated with the mProphet approach and set to 0.01 at both peptide precursor level and protein level (*Reiter et al., 2011*; *Rosenberger et al., 2017*). Due to the small sample size, for this dataset, we considered the top 1000 protein correlates of EAA (|r| = 0.25, p=0.06 for the pan-tissue EAA; |r| = 0.31, p=0.02 for the liver int.EAA).

## Data availability

The normalized microarray data and raw files are available from the NCBI Gene Expression Omnibus (accession ID GSE199979). The HorvathMammalMethylChip40 array manifest files and genome annotations of CpGs can be found on GitHub at https://github.com/shorvath/MammalianMethylationConsortium (*Horvath and Haghani, 2021*). Individual-level BXD data, including the processed microarray

data, are available on https://www.genenetwork.org (*Williams, 2022*) in FAIR+-compliant format; data identifiers and way to retrieve data are described in *Supplementary file 13*.

## Acknowledgements

We thank the entire UTHSC BXD Aging Colony team, particularly Dr. Suheeta Roy, Casey J Chapman, Melinda S McCarty, Jesse Ingles, and everyone else who contributed to the tissue harvest. We thank Dr. Lu Lu, who leads the main BXD Colony effort. We thank Dr. Evan G Williams for making the gene expression data readily available, and Dr. David Ashbrook for providing the BXD genotypes. We thank the GeneNetwork team, especially Zach Sloan and Arthur Centeno, who have been extremely prompt and effective at assisting with the GeneNetwork interface, and Dr. Pjotr Prins for his role in implementing GEMMA. We thank Dr. Garrett Jenkinson for the invaluable guidance he provided for entropy estimation. This study was funded by the NIH NIA grants R21AG055841 and R01AG043930.

## Additional information

### Competing interests

Steve Horvath: is a founder of the non-profit Epigenetic Clock Development Foundation, which plans to license several of his patents from his employer, University of California Regents. The Regents of the University of California filed a patent application (publication number WO2020150705) related to the HorvathMammalMethylChip40 and clock computation for which he is named an inventor. The other authors declare that no competing interests exist.

### Funding

| Funder | Grant reference number | Author |
|---|---|---|
| National Institute on Aging | R21AG055841 | Khyobeni Mozhui |
| National Institute on Aging | R01AG043930 | Robert W Williams |

The funders had no role in study design, data collection and interpretation, or the decision to submit the work for publication.

### Author contributions

Khyobeni Mozhui, Conceptualization, Data curation, Formal analysis, Funding acquisition, Investigation, Methodology, Visualization, Writing - original draft, Writing – review and editing; Ake T Lu, Caesar Z Li, Amin Haghani, Formal analysis, Methodology, Writing – review and editing; Jose Vladimir Sandoval-Sierra, Writing – review and editing, laboratory work, contributed to data generation; Yibo Wu, Methodology, Resources, Writing – review and editing; Robert W Williams, Conceptualization, Data curation, Resources, Writing – review and editing; Steve Horvath, Conceptualization, Data curation, Formal analysis, Methodology, Writing – review and editing

### Author ORCIDs

Khyobeni Mozhui http://orcid.org/0000-0002-6623-4112
Amin Haghani http://orcid.org/0000-0002-6052-8793
Jose Vladimir Sandoval-Sierra http://orcid.org/0000-0003-2007-6582

### Ethics

All animal procedures were in accordance to protocol approved by the Institutional Animal Care and Use Committee (IACUC) at the University of Tennessee Health Science Center. Protocol numbers 12-148.0 (2012-2015), 15-124.0 (2015-2018), and 18-094.0 (2018-present).

### Decision letter and Author response

Decision letter https://doi.org/10.7554/eLife.75244.sa1
Author response https://doi.org/10.7554/eLife.75244.sa2

## Additional files

### Supplementary files

- Supplementary file 1. Individual-level sample information.
- Supplementary file 2. CpGs and coefficients for the mouse clocks.
- Supplementary file 3. Covariates of the DNA methylation-based readouts.
- Supplementary file 4. Supplementary Tables 4a–c. (a) Sex differences in epigenetic aging after correction for body weight. (b) Pearson correlations between epigenetic age acceleration and strain-level longevity summaries. (c) High-priority candidate genes in quantitative trait locus for epigenetic age acceleration.
- Supplementary file 5. Multivariable regression analysis of epigenetic age acceleration.
- Supplementary file 6. Epigenome-wide association study results and annotations for 27,996 CpG probes.
- Supplementary file 7. Multivariable regression analysis of entropy by age-effect.
- Supplementary file 8. Genomic regions enrichments analyses for the differentially methylated CpGs.
- Supplementary file 9. Enrichment/depletion in genomic regions and chromatin states for the differentially methylated CpGs.
- Supplementary file 10. Quantitative trait locus analysis of epigenetic age acceleration and methylome-wide entropy.
- Supplementary file 11. Positional candidate genes in Eaa11 and Eaa19.
- Supplementary file 12. Transcriptome and proteome analysis. (a) Top 2000 liver transcriptome-wide correlates of liver int.EAA and pan-tissue EAA. (b) Functional enrichment among gene expression correlates of pan-tissue general clock (pan-tissue EAA). (c) Functional enrichment among gene expression correlates of liver interventional clock (liver int.EAA). (d) Top 2000 liver proteome correlates of liver int.EAA and pan-tissue EAA. EAA, epigenetic age acceleration.
- Supplementary file 13. Data access.
- Transparent reporting form

### Data availability

The normalized microarray data and raw files are available from the NCBI Gene Expression Omnibus (accession ID GSE199979). The HorvathMammalMethylChip40 array manifest files and genome annotations of CpGs can be found on Github at (https://github.com/shorvath/MammalianMethylationCons ortium; copy archived at swh:1:rev:355098ec161ae668546a1a343e755c40a475b941). Individual level BXD data, including the processed microarray data are available on http://www.genenetwork.org/ on FAIR+ compliant format; data identifiers, and way to retrieve data are described in Supplementary file 13.

The following datasets were generated:

| Author(s) | Year | Dataset title | Dataset URL | Database and Identifier |
|---|---|---|---|---|
| Mozhui K, Lu AT, Li CZ, Haghani A, Sandoval-Sierra JV, Wu Y, Williams RW, Horvath S | 2022 | Genetic Loci and Metabolic States Associated With Murine Epigenetic Aging | https://www.ncbi. nlm.nih.gov/geo/ query/acc.cgi?acc= GSE199979 | NCBI Gene Expression Omnibus, GSE199979 |
| Longevityteam | 2021 | Genetics and epigenetics of aging and longevity in BXD mice | http://www. genenetwork.org/ show_trait?trait_id= 10071&dataset=BXD-LongevityPublish | BDL_10071, 10071 |

*Continued on next page*

*Continued*

| Author(s) | Year | Dataset title | Dataset URL | Database and Identifier |
|---|---|---|---|---|
| Longevityteam | 2021 | Genetics and epigenetics of aging and longevity in BXD mice | http://www.genenetwork.org/show_trait?trait_id=10072&dataset=BXD-LongevityPublish | BDL_10072, 10072 |
| Longevityteam | 2021 | Genetics and epigenetics of aging and longevity in BXD mice | http://www.genenetwork.org/show_trait?trait_id=10073&dataset=BXD-LongevityPublish | BDL_10073, 10073 |
| Longevityteam | 2021 | Genetics and epigenetics of aging and longevity in BXD mice | http://www.genenetwork.org/show_trait?trait_id=10074&dataset=BXD-LongevityPublish | BDL_10074, 10074 |
| Longevityteam | 2021 | Genetics and epigenetics of aging and longevity in BXD mice | http://www.genenetwork.org/show_trait?trait_id=10075&dataset=BXD-LongevityPublish | BDL_10075, 10075 |
| Longevityteam | 2021 | Genetics and epigenetics of aging and longevity in BXD mice | http://www.genenetwork.org/show_trait?trait_id=10076&dataset=BXD-LongevityPublish | BDL_10076, 10076 |
| Longevityteam | 2022 | Genetics and epigenetics of aging and longevity in BXD mice | http://www.genenetwork.org/show_trait?trait_id=10093&dataset=BXD-LongevityPublish | BDL_10093, 10093 |

The following previously published datasets were used:

| Author(s) | Year | Dataset title | Dataset URL | Database and Identifier |
|---|---|---|---|---|
| Longevityteam | 2021 | Genetics of longevity in BXD mice | http://www.genenetwork.org/show_trait?trait_id=10001&dataset=BXD-LongevityPublish | BDL_10001, 10001 |
| Longevityteam | 2021 | Genetics of longevity in BXD mice | http://www.genenetwork.org/show_trait?trait_id=10002&dataset=BXD-LongevityPublish | BDL_10002, 10002 |
| Longevityteam | 2021 | Genetics of longevity in BXD mice | http://www.genenetwork.org/show_trait?trait_id=10003&dataset=BXD-LongevityPublish | BDL_10003, 10003 |
| Longevityteam | 2021 | Genetics of longevity in BXD mice | http://www.genenetwork.org/show_trait?trait_id=10004&dataset=BXD-LongevityPublish | BDL_10004, 10004 |
| Longevityteam | 2021 | Genetics of longevity in BXD mice | http://www.genenetwork.org/show_trait?trait_id=10005&dataset=BXD-LongevityPublish | BDL_10005, 10005 |

*Continued on next page*

*Continued*

| Author(s) | Year | Dataset title | Dataset URL | Database and Identifier |
|---|---|---|---|---|
| Longevityteam | 2021 | Genetics of longevity in BXD mice | http://www.genenetwork.org/show_trait?trait_id=10006&dataset=BXD-LongevityPublish | BDL_10006, 10006 |
| Longevityteam | 2021 | Genetics of longevity in BXD mice | http://www.genenetwork.org/show_trait?trait_id=10010&dataset=BXD-LongevityPublish | BDL_10010, 10010 |
| Longevityteam | 2021 | Genetics of longevity in BXD mice | http://www.genenetwork.org/show_trait?trait_id=10011&dataset=BXD-LongevityPublish | BDL_10011, 10011 |
| Longevityteam | 2020 | Genetics of longevity in BXD mice | http://www.genenetwork.org/show_trait?trait_id=10021&dataset=BXD-LongevityPublish | BDL_10021, 10021 |
| Longevityteam | 2020 | Genetics of longevity in BXD mice | http://www.genenetwork.org/show_trait?trait_id=10022&dataset=BXD-LongevityPublish | BDL_10022, 10022 |
| Longevityteam | 2020 | Genetics of longevity in BXD mice | http://www.genenetwork.org/show_trait?trait_id=10025&dataset=BXD-LongevityPublish | BDL_10025, 10025 |
| Longevityteam | 2021 | Genetics and epigenetics of aging and longevity in BXD mice | http://www.genenetwork.org/show_trait?trait_id=10066&dataset=BXD-LongevityPublish | BDL_10066, 10066 |

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
