## [Editor Report]

This article used three newly generated epigenetic predictors to test how they differ between genetically diverse mice from the BXD family (by looking at metabolic traits and lifespan). The authors subsequently identified several quantitative trait loci for the different predictors, using linkage analysis, and performed transcriptome and proteome analyses of liver and adipose tissue. The described results provide some important new insights on the underlying biology of epigenetic mouse aging and may be used to inform future studies in other model organisms and humans focused on studying the relationship between epigenetic aging and metabolism.

---

## [Decision Letter]

**Decision letter after peer review:**

Thank you for submitting your article "Genetic Analyses of Epigenetic Predictors that Estimate Aging, Metabolic Traits, and Lifespan" for consideration by *eLife*. Your article has been reviewed by 3 peer reviewers, and the evaluation has been overseen by a Reviewing Editor and Carlos Isales as the Senior Editor. The following individual involved in review of your submission has agreed to reveal their identity: Ferdinand von Meyenn (Reviewer #3).

Essential revisions:

1) The authors make use of multiple epigenetic predictors, but as indicated by Reviewer 2, some of the previously published clocks have not been taken into account. Moreover, as mentioned by Reviewer 1, most of the clocks have only been published in preprints that have not yet been peer reviewed. Hence, the questions is if these clocks will still be the same after peer review (given that they likely do not meet all the criteria for good reproducibility, as outlined here https://www.biorxiv.org/content/10.1101/2021.04.16.440205v1). The fact that the different clocks show slightly different results should also be discussed in more detail. The manuscript also has a large overlap with a previous published paper by the same authors (PMID: 32790008), which limits the novelty of the current manuscript.

2) The reviewers were intrigued by the findings regarding the Shannon entropy. However, the effects are very small, and the question is how meaningful the differences actually are. As shown in the "Recommendations for the authors" sections the reviewers suggest some additional analyses to further confirm these findings.

3) Some of the findings may be biased because the authors only used transcriptomic and proteomic data from the liver. The question is thus how generalizable the findings are across tissues. The transcriptomic and proteomic analyses are overall very superficial and should be extended and discussed in more detail.

The authors should also address the additional points mentioned in the individual review reports below.

*Reviewer #1 (Recommendations for the authors):*

This reviewer finds the observation that metabolic state is inducing changes in epigenetic age very interesting and worth studying. Series of correlations with acceleration of epigenetic age is well presented and quantified. However, some observations, although in lower numbers, were previously presented by the laboratory in another paper.

Description of QTLs is one of the most interesting parts of the study. Correlations of metabolically relevant loci with age acceleration is highly suggestive of molecular mechanism, probably based on negative feedback loop, that moderates DNAm. This idea is however not discussed in the paper.

Interesting questions arise – DNAm entropy hypothesis implies that most of the DNA methylation changes with age is, to certain degree, random, and as such has no function. Yet in the same study, authors define interesting correlations between the DNAm prediction and metabolism and other traits. How does it correlate with entropy theory? Authors are definitely surprised but they miss opportunity to discuss possibility that depending on the tissue some markers can have function. Are there markers specifically affected in given tissue that correlate with given phenotype? Or there is another reason that authors would like to elaborate on. Once the metabolism is discussed we need to take into account tissue-tissue physiological interactions. This seems to surprise authors that postulate diet induced changes in age based on DNA methylation affecting genes involved in the given metabolic pathway.

The last part of the study – QTLs and genes associated with epigenetic age acceleration is extremely interesting. Many of these genes are involved either in DNA methylation itself, DNA methylation reading or lipid metabolism, lipid transport, lipid oxidation, cholesterol (another lipid) metabolism, and finally response to oxidative stress, inflammation etc. This list of genes is also seemingly at odds with entropy theory if one takes into account series negative feedback loops in many of these pathways.

Although interesting and relevant for the field, this reviewer does not find this study extremely novel. The data presented in part confirm the data presented previously by the same laboratory as well as by others and addition of QTLs (novel idea, stemming from genetic variants field) is not developed enough to warrant publication in the premier journal.

Probably the highest weakness of the study is the fact that this work is part of the thread of manuscripts deposited recently in biorxiv by the same laboratory. This reviewer finds this surprising and quite unnecessary to split several works into many manuscripts. In particular, the generation of the pan-mammalian epigenetic clock, separately presented in biorxiv, is at the base of this work. Since other studies are not peer-reviewed authors assume that reviewers/readers will "believe" all data presented in first part of the study. These studies should not be published separately. Unless the pan-mammalian clock work is planned to be published back-to-back with this study?

*Reviewer #2 (Recommendations for the authors):*

1. Page 3. It will be more beneficial to the broad readership if the authors give a more thorough introduction of the current state of epigenetic clocks based on their expertise. Several manuscripts can be cited to improve the introduction, such as clocks built on mouse samples (Thompson et al., 2018; Wang and Lemos, 2019).

2. Page 4 Table 1. Additional universal clocks were described by the authors (Lu et al., 2021), including the log-linear age clock. I am curious why the authors only provided the relative age clock in this particular study. Do both universal clocks show similar results?

3. Page 6. The results from Table 2 showed that clocks built from different CpG subsets or trained on different samples tend to tell different results in terms of significance, in particular when testing correlations between EAA and BW0 in the chow diet groups. I wonder if the authors could give a more detailed interpretation of these differences. Does it mean the correlation exists, but some clocks are just not as precise as others in catching it?

4. Page 6. The finding that Shannon entropy had a (weak) significant negative correlation with weight in chow diet mice is interesting. Previous work on Shannon entropy and aging in mice described a method of calculating the entropy effect of CpG sites with increased and decreased methylation with age separately (Sziraki et al., 2018). Will the sites that increase and decrease methylation with age show different correlations with body weight?

5. Page 13. Interestingly, the cytochrome P450 cluster was identified in Eaaq 19 when the authors try to identify candidate genes for epigenetic age acceleration, as the function of P450 is typically unique to the liver. Would the authors comment on the potential "universal" mechanisms that may drive epigenetic age acceleration across tissues and organs? Or do the authors think that the epigenetic aging trajectories are unique in every organ?

6. References for Table S4 has an invalid citation (number 3).

References:

Lu, A.T., Fei, Z., Haghani, A., Robeck, T.R., Zoller, J.A., Li, C.Z., Zhang, J., Ablaeva, J., Adams, D.M., Almunia, J., et al. (2021). Universal DNA methylation age across mammalian tissues. bioRxiv, 2021.2001.2018.426733.

Sziraki, A., Tyshkovskiy, A., and Gladyshev, V.N. (2018). Global remodeling of the mouse DNA methylome during aging and in response to calorie restriction. Aging Cell 17, e12738.

Thompson, M.J., Chwialkowska, K., Rubbi, L., Lusis, A.J., Davis, R.C., Srivastava, A., Korstanje, R., Churchill, G.A., Horvath, S., and Pellegrini, M. (2018). A multi-tissue full lifespan epigenetic clock for mice. Aging 10, 2832-2854.

Wang, M., and Lemos, B. (2019). Ribosomal DNA harbors an evolutionarily conserved clock of biological aging. Genome Research 29, 325-333.

*Reviewer #3 (Recommendations for the authors):*

This study is a very nice and comprehensive analysis of genetic and environmental factors affecting DNA methylation during ageing. Overall, the major addition I could see to significantly change the manuscript would be targeted genetic manipulation of either the identified genomic regions or the DNA methylation machinery. Otherwise, I have only few suggestions/comments beyond the ones mentioned above:

The study combines a number of research areas which may otherwise not interact regularly and uses terminology and methods from each of these. I might be helpful to include some reference values for the interpretation of the magnitude of effects and also some guidance in assessing the individual contribution of the various factors study on epigenetic age. I think this would help the broader audience to better understand which effects are highly relevant and which may be only weak and to be taken with care (due to sample size or effect size).

The final comparison with proteomics and transcriptomic data is also very brief – maybe a comment as to whether no strong hits were found or whether the analysis is not extensive (which could then be done?) would also add value to this part.

The authors could briefly comment on the prior published results on effects of HFD on DNAm age.

---

## [Author Response]

Essential revisions:1) Inclusion of clocks that are in preprint stage:

Although the results are very robust, those pan-mammal predictors have been excluded as the referenced papers are still under review. (See point 5 in response to Reviewer 1)

2) “… different clocks show slightly different results should also be discussed…”:

A more thorough comparison between the clocks is now included. (See point 3 in response to Reviewer 2).

3) “The manuscript also has a large overlap with a previous published paper… which limits the novelty of the current manuscript.”:

We highlight the overlaps, but we have novel results that are unique to the present work (see point 1 in response to Reviewer 1).

4) “The reviewers were intrigued by the findings regarding the Shannon entropy… suggest some additional analyses…”:

We have carried out the recommended analyses, and frankly, very thankful to the reviewer for making this very thoughtful suggestion. (see point 4 under Reviewer 2)

5) “… findings may be biased because the authors only used transcriptomic and proteomic data from the liver… transcriptomic and proteomic analyses are overall very superficial and should be extended and discussed in more detail.”:

We have tried to address this in the revised manuscript, and now include proteome data from adipose tissue. We have also elaborated on the tissue-dependency in the Discussion section (see point 4 under Reviewer 1, and point 5 under Reviewer 2).

Reviewer #1 (Recommendations for the authors):1) “… some observations, although in lower numbers, were previously presented…”

We highlight that this is an extension of our previous work in the introduction. Pg4: “In our previous work… reported rapid age-dependent methylation changes in mice on high fat diet, and in mice with higher body weight”.

To the best of our knowledge, the present work is the first genetic mapping study of epigenetic age acceleration in mice, and provides evidence that entropy may contribute to these clock readouts, and is coupled to the metabolic state. This is the novel part of the work.

2) “… molecular mechanism, probably based on negative feedback loop, that moderates DNAm…”

We thank the reviewer for this insightful comment, and for pointing out that we need a fuller discussion on this. We have now gone into greater detail.

“The centrality of bioenergetics for biological systems may explain why we detect this coupling between the DNAm readouts (i.e., the clocks, and entropy), and indices of metabolism including weight, diet, levels of macronutrients, and even expression of metabolic genes… many metabolites (e.g., SAM, NAD^+^, ATP) are essential co-factors for enzymes that shape the epigenome, and these could serve as nutrient sensors and mechanistic intermediaries that regulate how the epigenome is organized in response to metabolic conditions. Close interactions between macro- and micronutrients, and DNAm is a conserved process…”

3) “… DNAm entropy hypothesis implies… random, and as such has no function… interesting correlations between the DNAm prediction and metabolism and other traits. How does it correlate with entropy theory?”

We thank the reviewer for the thought provoking comments on entropy. Kindly note that we have updated the entropy computation after consulting with Dr. Garrett Jenkinson, who has published some intriguing works on epigenetic entropy.

While entropy itself is an increase in randomness, the change from an ordered to a disordered state can result in loss of function. One of the most interesting discussions on this was by Hayflick, 2007 (we now cite this paper). This loss of maintenance and function may be particularly relevant for CpGs that are highly conserved, and presumed to have important functions. Furthermore, the *rate* at which error accumulates may not be entirely random and passive, and could be influenced by metabolic conditions and genetic factors.

How well and how long a biological system is able to maintain a more ordered state may be dependent on a number of factors. While entropy may partly contribute to the DNAm clocks, this is unlikely to be the only source of epigenetic aging, and the significant strain variation and the heritability (albeit, modest) suggests that there is a combination of processes, and also influenced by genetic variation.

We discuss this:

“… This leads us to suggest that the rate of noise accumulation, an aspect of epigenomic aging, can vary between individuals, and the resilience or susceptibility to this shift towards higher noise may be partly modulate by diet as well as genetic factors.”

4) “… they miss opportunity to discuss possibility that depending on the tissue some markers can have function. Are there markers specifically affected in given…”

We have now included more results and discussion on the tissue effect. In addition to the liver transcriptome and proteome, we performed analyses with adipose proteome. These new results can be found under the Results section “Gene expression correlates of EAA” pg 17, and in pg 21 of discussion.

“…depending on the tissue(s) in which the clocks are trains, and the tissue from which the DNAmAge is estimated, the EAA derivative may put an emphasis on biological pathways or genes that are most relevant to that tissue… With the liver clocks, expression correlates highlighted aspects of metabolism that are relevant to liver function… For the adipose tissue proteome, the cytochrome genes become less prominent, but the enriched pathways still remained consistent…”

5) “… weakness of the study is… part of the thread of manuscripts deposited recently in biorxiv… the generation of the pan-mammalian epigenetic clock… Since other studies are not peer-reviewed authors assume that reviewers/readers will "believe" all data presented in first part of the study…”

We agree with the reviewer. The timing did not quite align. We are confident that the results we see are robust. However, since the pan-mammalian clocks and lifespan predictor are not officially published, we have excluded those, and now only use the mouse clocks, which are specific to this study. There are now a number of such clocks, and building clocks have become fairly standard. The primary goal of this manuscript was to examine what biological variable associated with these clocks. Excluding the pan-mammal predictors do not change the overall results.

6) “… proteomic and transcriptomic study was performed in liver…”

We have included adipose tissue, and have also elaborated on this in the discussion (see response under point 4)

Reviewer #2 (Recommendations for the authors):1) “… more beneficial to the broad readership if the authors give a more thorough introduction of the current state of epigenetic clocks…”

We have expanded on this in the Introduction and have included additional references

“… many different models of human DNAm clock have been develop, and this rapid expansion was made possible by reliable DNAm microarrays that provide a fixed CpG content… performance of these clocks depend heavily on the training models, and the size and tissue types of the training set...”

“… extended to model organisms, and this has opened up the possibility of directly testing the effects of different interventions… Most rodent studies have used enrichment-based DNAm sequencing, and this limits the transferability and reproducibility…”

Additional discussions have also been added:

“…well-known that DNAm clocks have high level of degeneracy… highly accurate predictors of chronological age can be built from entirely different sets of CpGs… all these DNAm clocks achieve reasonably high prediction of chronological age, the age divergence derived from these different clocks (EAA) can capture slightly different facets of biological aging, and the better a clock is at predicting chronological age, the lower its association with mortality risk.[ref]”

2) “… additional universal clocks were described by the authors (Lu et al., 2021)… Do both universal clocks show similar results?”

The results are consistent using both the universal clocks. However, we have now excluded the pan-mammalian predictors as the referenced manuscripts are in preprint stage. We only use the mouse clocks, which are specific to this study, and excluding the universal predictors does not change the overall results.

3) “… clocks built from different CpG subsets or trained on different samples tend to tell different results… if the authors could give a more detailed interpretation…”

Prompted by this comment, we have now added new figures (Figure 1h, Figure S1) to contrast between the clocks, and have expanded on it in the Discussion. We thank the reviewer for this improvement.

“…we find that the interventional clocks deviated most from chronological age… interventional clocks were also associated with BWF and cholesterol, but had weaker associations with BW0… was the clock that had the strongest inverse correlation with strain longevity. In contrast, the developmental clocks, which were based on CpGs that change early in life, showed a stronger association with BW0…”

4) “Previous work on Shannon entropy and aging in mice described a method of calculating the entropy effect of CpG sites with increased and decreased methylation with age separately (Sziraki et al., 2018). Will the sites that increase and decrease methylation with age show different correlations with body weight?”

Once again, we thank the reviewer for this very helpful suggestion. (Kindly note that we have updated the entropy computation after consulting with Dr. Garrett Jenkinson.)

We have added a new section in the Results (*“Multifactor variance of the conserved CpGs”*) where we identify the direction of change of CpGs with age (Figure 2), and examine the entropy changes in a manner similar to what is reported by Sziráki (Figure 3). We make sure to acknowledge that this was inspired by Sziráki et al.

These additional analyses have uncovered intriguing findings, and we are very glad that the reviewer pointed us in that direction.

5) “Would the authors comment on the potential "universal" mechanisms that may drive epigenetic age acceleration across tissues and organs? Or do the authors think that the epigenetic aging trajectories are unique in every organ?”

We have now included more results and discussion on the tissue effect. In addition to the liver transcriptome and proteome, we performed analyses with adipose proteome. These new results can be found under the Results section “*Gene expression correlates of EAA*” pg 17, and in pg 21 of discussion.

“…depending on the tissue(s) in which the clocks are trains, and the tissue from which the DNAmAge is estimated, the EAA derivative may put an emphasis on biological pathways or genes that are most relevant to that tissue… With the liver clocks, expression correlates highlighted aspects of metabolism that are relevant to liver function… For the adipose tissue proteome, the cytochrome genes become less prominent, but the enriched pathways still remained consistent…”

6) References for Table S4 has an invalid citation (number 3).

We thank the reviewer for the detailed reading of our manuscript. We have corrected this error.

Reviewer #3 (Recommendations for the authors):1) “…comprehensive analysis of genetic and environmental factors.. the major addition I could see to significantly change the manuscript would be targeted genetic manipulation…”

We thank the reviewer for their commendation. Targeted genetic manipulation is an important next step that we have considered. Unfortunately, we currently do not have the resources to carry out such a study, but we are eager to do so, and in the process of submitting grant proposals.

In the discussion, we highlight this weakness.

“… exemplify the major challenge that follows when a genetic mapping approach leads to gene- and variant-dense regions. [ref] Both loci have several biologically relevant genes, and identifying the causal gene (or genes) will require a more fine-scaled functional genomic dissection.”

2) “… helpful to include some reference values for the interpretation of the magnitude of effects and also some guidance in assessing the individual contribution of the various factors…”

This is a very helpful input that will enhance the manuscript. We have updated the manuscript accordingly.

We now include additional information that will give a clearer idea of magnitude of effects throughout the manuscript.

E.g.: “…age accounted for about 6% (in CD) to 28% (in HFD) of the variance in entropy”… “… CD mice had an average of –0.04 years of age deceleration, and HFD mice had an average of +0.11 years of age acceleration… ”

We have also updated the figures and provide R-squared values (variance explained) wherever relevant.

3) “The final comparison with proteomics and transcriptomic data is also very brief…”

Based on the reviewer’s comment, we have done additional analyses and added more detail on the proteomic and transcriptomic analyses. The relevant figure (Figure 6) has also been updated. Instead of one clock, we now examine the transcripts and proteins (and pathways) that are the most consistent or different between two clocks (the general pan-tissue clock, and the liver-specific intervention clock). We also included a similar analysis using proteome data from adipose tissue.

4) “The authors could briefly comment on the prior published results on effects of HFD on DNAm age.”

We have updated this information in the Discussion section.

Pg19: “HFD had an age accelerating effect on the clocks, and this is concordant with our previous report where we found more rapid age-associated changes in methylation. [ref] This also concurs with studies in humans that have found that obesity accelerates epigenetic aging.[Ref]”

5) Figure 3: labels are off, ie no "d". Also, Figure 3c is not referenced in the text.

We are very thankful to the reviewer for the careful evaluation of our manuscript and for highlighting these errors. This figure has been replaced.